# Understanding and Improving Fast Adversarial Training against $l_0$ Bounded Perturbations

**Xuyang Zhong**
Department of Computer Science
City University of Hong Kong
xuyang.zhong@my.cityu.edu.hk

**Yixiao Huang**
Department of Computer Science
City University of Hong Kong
yixiao.huang@my.cityu.edu.hk

**Chen Liu** *
Department of Computer Science
City University of Hong Kong
chen.liu@cityu.edu.hk

## Abstract

This work studies fast adversarial training against sparse adversarial perturbations bounded by $l_0$ norm. We first demonstrate the unique challenges of employing 1-step attacks on $l_0$ bounded perturbations, especially catastrophic overfitting (CO) that cannnot be properly addressed by existing fast adversarial training method for other $l_p$ norms ($p \geq 1$). We highlight that CO in $l_0$ adversarial training arises from sub-optimal perturbation locations of 1-step attack. Some strategies like multi-$\epsilon$ can mitigate this sub-optimality to some extent, they lead to unstable training in turn. Theoretical and numerical analyses also reveal that the loss landscape of $l_0$ adversarial training is more craggy than its $l_\infty$, $l_2$ and $l_1$ counterparts, which exaggerates CO. To address this issue, we adopt soft labels and the trade-off loss function to smooth the adversarial loss landscape. Extensive experiments demonstrate our method can overcome the challenge of CO, achieve state-of-the-art performance, and narrow the performance gap between 1-step and multi-step adversarial training against sparse attacks. Codes are available at https://github.com/CityU-MLO/sPGD.

## 1 Introduction

Deep neural networks have been shown vulnerable to adversarial perturbations [1]. To achieve robust models, comprehensive evaluations [2, 3, 4] have demonstrated that adversarial training [5] and its variants [6, 7, 8, 9, 10, 11, 12] are the most effective methods. However, adversarial training is generally computationally expensive because generating adversarial perturbations in each training step needs multiple forward and backward passes of the model. Such efficiency issues hinder the scalability of adversarial training.

Improving the efficiency of adversarial training is tricky. Some works [13, 14, 15, 16] employ faster but weaker 1-step attacks to generate adversarial perturbations for training. However, such methods may suffer from *catastrophic overfitting (CO)* [17]: the model overfits these weak attacks instead of achieving true robustness against adaptive and stronger attacks. On the other hand, most existing works [5, 18, 19] focus on studying adversarial perturbations bounded by $l_\infty$, $l_2$ or $l_1$ norms. In these scenarios, the set of allowable perturbations is *convex*, which facilitates optimizing adversarial perturbations and thus adversarial training. However, there are many scenarios in real-

---

*Corresponding Author

39th Conference on Neural Information Processing Systems (NeurIPS 2025).

Table 1: Robust accuracy of sAT and sTRADES [23] with different steps ($t$) evaluated by Sparse-AutoAttack (sAA) [23], where the sparsity level is $\epsilon = 20$. The models are PreactResNet-18 [24] trained on CIFAR-10 [25].

|  | sAT ($t = 1$) | sAT ($t = 20$) | sTRADES ($t = 1$) | sTRADES ($t = 20$) |
|---|---|---|---|---|
| Robust Acc. | 0.0 | 36.2 | 31.0 | 61.7 |

world applications where sparse perturbations, bounded by the $l_0$ norm, need to be considered [20, 21, 22, 23]. Since the $l_0$ norm is not a proper norm, the set of all allowable perturbations in this case is *not convex*. Consequently, from an optimization perspective, obtaining robust models against sparse perturbations becomes more challenging. Compared with the $l_\infty$, $l_2$ and $l_1$ counterparts, more steps are needed to generate strong $l_0$ bounded perturbations, making the corresponding adversarial training even more computationally expensive.

Among algorithms aiming at obtaining robust models against sparse perturbations, sAT and sTRADES [23] stand out as the most effective ones. These methods employ adversarial training against Sparse-PGD (sPGD) [23]. However, they still require 20 steps to generate adversarial perturbations in each training step to achieve decent performance. As demonstrated in Table 1, naively decreasing the number of steps to 1 leads to a significant performance decline for both sAT and sTRADES.

In this work, we investigate the challenges associated with fast adversarial training against sparse perturbations, including training instability caused by CO and performance decline in both robust and clean accuracy. Specifically, we highlight that CO in $l_0$ adversarial training is caused by sub-optimal perturbation locations of 1-step attack. Our observation indicates that adjusting the perturbation magnitudes alone cannot help mitigate CO in this context, so **many existing CO mitigation methods [26, 27, 28, 29] used in other cases do not work at all in the $l_0$ scenario**. Although the multi-$\epsilon$ strategy [19, 23] can mitigate sub-optimal perturbation locations, it suffers from unstable training and degraded clean accuracy. In light of these findings, we present empirical and theoretical evidence to illustrate that the loss landscape of adversarial training against $l_0$ bounded perturbations is markedly more craggy compared to its $l_\infty$, $l_2$, and $l_1$ counterparts. Furthermore, we corroborate that **the craggy loss landscape aggravates CO in $l_0$ adversarial training**.

Drawing from these insights, we adopt soft labels and a trade-off loss function to enhance the smoothness of the adversarial loss objective function, thereby improving the performance of fast adversarial training against sparse perturbations. Although similar smoothing techniques have been applied in adversarial training [30, 31, 32, 33], existing literature employs them to address robust overfitting to boost performance in $l_2$ and $l_\infty$ cases. By contrast, we demonstrate that it is essential and has a much larger performance improvement in the $l_0$ case of fast adversarial training. Finally, our extensive experiments demonstrate that smoothing the loss landscape can eliminate CO in the $l_0$ case, and significantly narrow the performance gap between 1-step adversarial training and its multi-step counterparts.

To the best of our knowledge, this work is the first to investigate fast adversarial training in the context of $l_0$ bounded perturbations. We summarize the contributions of this paper as follows:

1. We highlight that catastrophic overfitting (CO) in fast $l_0$ adversarial training arises from sub-optimal perturbation locations in 1-step attacks. Techniques effective in $l_\infty$, $l_2$, and $l_1$ cases cannot fully address the CO issue in the $l_0$ case.

2. Our theoretical and empirical analysis shows that the adversarial loss landscape is more craggy in $l_0$ cases, exacerbating CO. In this regard, we can adopt soft labels and the trade-off loss function to provably smooth the adversarial loss landscape.

3. Comprehensive experiments demonstrate that smoothing the loss function significantly narrows the performance gap between 1-step $l_0$ adversarial training and its multi-step counterparts, achieving state-of-the-art performance in efficient adversarial training against sparse perturbations.

**Notation and Terminology** Consider a classification model $F(\boldsymbol{x}, \boldsymbol{\theta}) = \{f_i(\boldsymbol{x}, \boldsymbol{\theta})\}_{i=0}^{K-1}$, where $\boldsymbol{x} \in \mathbb{R}^d$ is the input, $\boldsymbol{\theta}$ represents the parameters of the model, and $K$ is the number of classes, $f_i(\boldsymbol{x}, \boldsymbol{\theta})$ is the logit of the $i$-th class. Correspondingly, we use $\{h_i\}_{i=0}^{K-1}$ to represent the output probability of each class, which is the result of softmax function applied to $\{f_i\}_{i=0}^{K-1}$. Therefore, the

loss objective function $\mathcal{L}$ based on the cross-entropy is calculated as follows:

$$\mathcal{L}(\boldsymbol{x}, \boldsymbol{\theta}) \stackrel{\text{def}}{=} -\sum_{i=0}^{K-1} y_i \log h_i(\boldsymbol{x}, \boldsymbol{\theta}) \stackrel{\text{def}}{=} -\sum_{i=0}^{K-1} y_i \log \frac{\exp(f_i(\boldsymbol{x}, \boldsymbol{\theta}))}{\sum_{j=0}^{K-1} \exp(f_j(\boldsymbol{x}, \boldsymbol{\theta}))} \tag{1}$$

where $\boldsymbol{y} = [y_1, y_2, ..., y_C]$ is the label of $\boldsymbol{x}$ in a simplex, i.e., $\sum_i y_i = 1$. In the context of adversarial perturbation, we use $\mathcal{S}_\epsilon^{(p)}(\boldsymbol{x}) \stackrel{\text{def}}{=} \{\boldsymbol{\delta} | \|\boldsymbol{\delta}\|_p \leq \epsilon, 0 \leq \boldsymbol{x} + \boldsymbol{\delta} \leq 1\}$ to represent the adversarial budget, i.e., the set of all allowable input perturbations for the input $\boldsymbol{x}$. The adversarial loss function is $\mathcal{L}_\epsilon^{(p)}(\boldsymbol{x}, \boldsymbol{\theta}) \stackrel{\text{def}}{=} \max_{\boldsymbol{\delta} \in \mathcal{S}_\epsilon^{(p)}(\boldsymbol{x})} \mathcal{L}(\boldsymbol{x} + \boldsymbol{\delta}, \boldsymbol{\theta})$. Despite no guarantee to obtain the optimal perturbation in practice, to simplify the notation, we denote the term $\mathcal{L}_\epsilon^{(p)}$ also as the adversarial loss induced by the actual attack algorithms and omit the superscript $(p)$ when there is no ambiguity.

## 2 Related Works

**Adversarial Attacks:** The existence of adversarial examples is first identified in [1], which focuses on $l_2$ norm-bounded adversarial perturbations. Fast gradient sign method (FGSM) [34] introduces an efficient approach by generating perturbations bounded by its $l_\infty$ norm in a single step. Furthermore, projected gradient descent (PGD) [5] extends and improves FGSM [35] by iterative updating and random initialization. In addition to these white-box attacks where the attackers have full access to the models, there are also several black-box attacks [36, 37] where the attackers' access is restricted. AutoAttack (AA) [3] is an ensemble of both white-box and black-box attacks to ensure a more reliable evaluation of model's robustness.

**Adversarial Training:** Adversarial training [5, 6, 7, 8, 9, 10, 11, 12] has emerged as a popular and reliable framework to obtain robust models [2, 3]. Under this framework, we first generate adversarial examples and update model parameters based on these examples in each mini-batch update. Different adversarial training variants, such as TRADES [30] and MART [32], may have different loss objective functions for generating adversarial examples and updating model parameters. Furthermore, compared with training on clean inputs, adversarial training is shown to suffer more from overfitting [38, 39]. In this regard, self-adaptive training (SAT) [40], which utilizes historical predictions as the soft label, has demonstrated its efficacy in improving the generalization.

**Sparse Perturbations:** Adversarial budget defined by $l_1$ norm is the tightest convex hull of the one defined by $l_0$ norm. In this context, SLIDE [18] extends PGD and employs $k$-coordinate ascent to generate $l_1$ bounded perturbations. Similarly, AutoAttack-$l_1$ (AA-$l_1$) [41] extends AA to the $l_1$ case. However, AA-$l_1$ is found to generate non-sparse perturbations that SLIDE fails to discover [19], indicating that $l_1$ bounded perturbations are not necessarily sparse. Therefore, we use $l_0$ norm to strictly enforce sparsity. It is challenging to optimize over an adversarial budget defined by $l_0$ norm, because of non-convex adversarial budgets. While naively applying PGD in this case turns out sub-optimal, there are several black-box attacks, including CornerSearch [21] and Sparse-RS [22], and white-box attacks, including Sparse Adversarial and Interpretable Attack Framework (SAIF) [42] and Sparse-PGD (sPGD) [23], which address the optimization challenge of finding $l_0$ bounded perturbations. Ultimately, Sparse-AutoAttack (sAA) [23], combining the most potent white-box and black-box attacks, emerges as the most powerful sparse attack.

**Fast Adversarial Training:** While effective, adversarial training is time-consuming due to the use of multi-step attacks. To reduce the computational overhead, some studies [13, 14] employ faster one-step attacks in adversarial training. However, the training based on these weaker attacks may suffer from catastrophic overfitting (CO) [17], where the model overfits to these weak attacks instead of achieving true robustness against a variety of attacks. CO is shown to arise from distorted decision boundary caused by *sub-optimal perturbation magnitudes* [26]. There are several methods proposed to mitigate CO, including aligning the gradients of clean and adversarial samples [27], adding stronger noise to clean sample [43], adaptive step size [29], regularizing abnormal adversarial samples [44], adding layer-wise weight perturbations [45], and penalizing logits discrepancy [31]. Furthermore, compared to its $l_2$ and $l_\infty$ counterparts, CO is caused by overfitting to sparse perturbations during $l_1$ adversarial training [19]. To address this issue, Fast-EG-$l_1$ [19] is introduced to generate $l_1$ bounded perturbations by Euclidean geometry instead of coordinate ascent. In this work, we investigate fast adversarial training against $l_0$ bounded perturbations.

# 3 Unique Challenges in Fast $l_0$ Adversarial Training

To obtain robust models against sparse perturbations, preliminary efforts use 20-step sPGD in adversarial training, which introduces significant computational overhead. To accelerate training, we explore using 1-step sPGD in adversarial training. However, as reported in Table 1, the models obtained in this way exhibit weak robustness against stronger sparse attacks, such as sAA. In this section, we study the underlying factors that make fast $l_0$ adversarial training challenging.

## 3.1 Catastrophic Overfitting in $l_0$ Adversarial Training

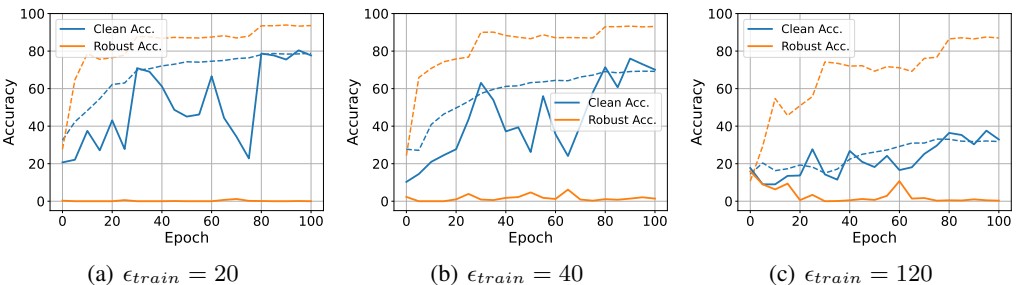

(a) $\epsilon_{train} = 20$      (b) $\epsilon_{train} = 40$      (c) $\epsilon_{train} = 120$

Figure 1: The learning curves of adversarial training against 1-step sPGD with random noise initialization. The models are PreactResNet-18 trained on CIFAR-10. The dashed and the solid lines represent the accuracy of the training and the test set, respectively. The test robust accuracy is based on sAA with $\epsilon = 20$. The values of $\epsilon$ used in training are shown as $\epsilon_{train}$, the training robust accuracy is based on the 1-step sPGD with $\epsilon_{train}$.

We plot the learning curves of adversarial training using 1-step sPGD in Figure 1. Specifically, we adopt the multi-$\epsilon$ strategy [19, 23] and allow for different adversarial budget sizes, i.e., $\epsilon$, during training and testing. The results in Figure 1 indicate that CO happens in all configurations. Moreover, our observations of CO in $l_0$ cases are different from other cases in several aspects. First, random initialization of adversarial perturbation, proven effective in $l_\infty$, $l_2$ and $l_1$ cases, does not yield similar results in the $l_0$ case. In addition, Figure 1 showcases that the training accuracy on the inputs perturbed by 1-step sPGD is even higher than their clean counterparts. What's more, when CO happens in $l_\infty$, $l_2$ and $l_1$ cases, the model sharply achieves perfect robustness against 1-step attacks but zero robustness against multi-step attacks, both in few mini-batch updates. Such phenomenon is not observed in $l_0$ cases. By contrast, we observe dramatic performance fluctuations on clean examples throughout the training process, even in the fine-tuning phase. Such training instability indicates a non-smooth landscape of the loss function in the parameter space: a subtle change in parameters $\boldsymbol{\theta}$ leads to abrupt fluctuation in the loss.

Moreover, we apply existing CO mitigation methods (ATTA [28], Free-AT [13], GradAlign (GA) [27], Fast-BAT [46], FLC Pool [47], N-AAER [44], N-LAP [45], and NuAT [16]), which are effective in $l_\infty$ and $l_2$ cases, in fast $l_0$ adversarial training. As shown in Table 2, **most methods designed for other $l_p$ norms ($p \geq 1$) turn out ineffective at all in the $l_0$ scenario**. Although NuAT achieves non-trivial performance, there is still a significant performance gap to multi-step methods. Therefore, it is imperative to ascertain more essential causes of CO in fast $l_0$ adversarial training and formulate effective mitigation strategies accordingly.

Table 2: Comparison between existing CO mitigation methods and multi-step method (sTRADES) in robust accuracy (%) by sAA. The target sparsity level $\epsilon = 20$. We compare PreAct ResNet-18 [24] models trained on CIFAR-10 [25]. The *italic numbers* indicate catastrophic overfitting (CO) happens. Note that all baselines are tuned by a thorough hyperparameter search.

| Method | ATTA | Free-AT | GA | Fast-BAT | FLC Pool | N-AAER | N-LAP | NuAT | sTRADES |
|---|---|---|---|---|---|---|---|---|---|
| Robust Acc. | *0.0* | 8.9 | *0.0* | 14.1 | *0.0* | *0.1* | *0.0* | 51.9 | 61.7 |

Table 3: Robust accuracy of the models obtained by 1-step sAT with different $\epsilon_{train}$ against the interpolation between perturbations generated by 1-step sPGD ($\epsilon = 20$) and their corresponding clean examples, where $\alpha$ denotes the interpolation factor, i.e., $\boldsymbol{x}_{interp} = \boldsymbol{x} + \alpha \cdot \boldsymbol{\delta}$. The results of sAA are also reported.

| $\alpha$ | 0.0 | 0.2 | 0.4 | 0.6 | 0.8 | 1.0 | **sAA** |
|---|---|---|---|---|---|---|---|
| $\epsilon_{train} = 20$ | 77.5 | **69.1** | 80.4 | 88.0 | 90.2 | 90.4 | **0.0** |
| $\epsilon_{train} = 40$ | 70.2 | **64.3** | 79.8 | 87.4 | 89.6 | 89.6 | **0.0** |
| $\epsilon_{train} = 120$ | 32.5 | **24.5** | 41.5 | 65.2 | 72.8 | 67.6 | **0.0** |

### 3.2 Unique Cause of CO in $l_0$ Adversarial Training: Sub-optimal Perturbation Location

In $l_\infty$ and $l_2$ cases, CO occurs due to distorted decision boundary caused by *sub-optimal perturbation magnitude* [26]. To ascertain if this applies to $l_0$ adversarial training, we evaluate the robustness accuracy of models trained by 1-step sAT with varying $\epsilon_{train}$ against interpolations between the clean inputs and the perturbed ones by 1-step sPGD. Table 3 shows that we cannot find successful adversarial examples through such simple interpolations.

By contrast, we notice that the adversarial perturbations generated by 1-step sPGD during training are almost completely different from those generated by sAA in location rather than magnitude (see in Figure 2(a)). Combining with the results in Table 3, we can demonstrate that *CO in $l_0$ adversarial training is primarily due to sub-optimal perturbation locations rather than magnitudes*.

Due to non-convexity and unique project operator to $l_0$ adversarial budget, most existing methods designed for $l_2$ or $l_\infty$ cases do not help improve the perturbation locations. Instead, we study multi-$\epsilon$ strategy, which is particularly effective in multi-step $l_0$ adversarial training [23]. Figure 2(b) indicates that the perturbations generated by 1-step attack with larger $\epsilon_{train}$ overlap more with those generated by sAA with a smaller and fixed $\epsilon_{test}$ in terms of location. **These findings suggest that the sub-optimal location issue brought by 1-step attacks can be mitigated to some extent by multi-$\epsilon$ strategy. However, as illustrated in Figure 1, a larger $\epsilon_{train}$, in turn, leads to unstable training and degraded clean accuracy.** To address this challenge, we investigate the loss landscape in the subsequent sections.

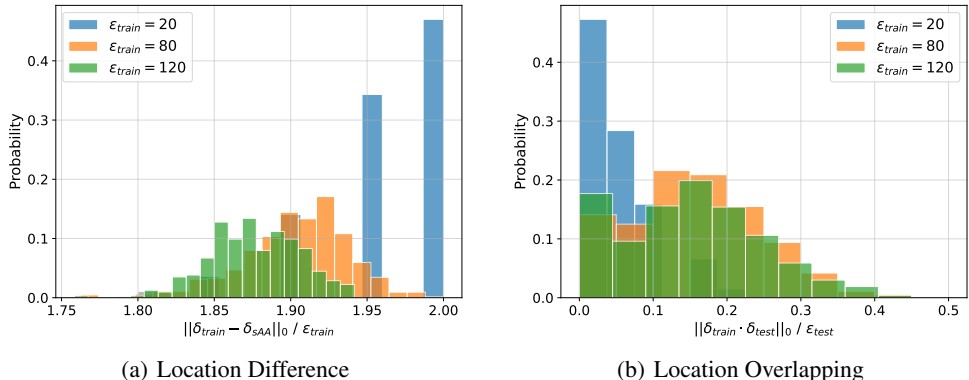

(a) Location Difference        (b) Location Overlapping

Figure 2: Visualization of location difference and location overlapping. **(a)** The distribution of the normalized $l_0$ distance between training adversarial examples generated by 1-step sPGD and sAA. The models trained on 20-step sAT with different training $\epsilon$ are evaluated. **(b)** The distribution of the location overlapping rate between the perturbations generated by attacks used in training (20-step sPGD) and test (sAA), where $\epsilon_{test} = 20$. The models trained on 20-step sAT with different training $\epsilon$ are evaluated.

## 4 Fast $l_0$ Adversarial Training Requires Loss Smoothing

In this section, we investigate the landscape of adversarial loss through theoretical analyses and numerical experiments. Ultimately, we provide our solution to stabilize and improve the performance of fast $l_0$ adversarial training.

## 4.1 Theoretical Analysis

We first provide theoretical analyses on the smoothness of adversarial loss function. Similar to [48], we assume the first-order smoothness of the model's outputs $\{f_i\}_{i=0}^{K-1}$.

**Assumption 4.1. (First-order Lipschitz condition)** $\forall i \in \{0, 1, ..., K-1\}$, the function $f_i$ satisfies the following first-order Lipschitz conditions, with constants $L_{\boldsymbol{\theta}}, L_{\boldsymbol{x}}$:

$$\forall \boldsymbol{x}, \boldsymbol{\theta}_1, \boldsymbol{\theta}_2, \ \|f_i(\boldsymbol{x}, \boldsymbol{\theta}_1) - f_i(\boldsymbol{x}, \boldsymbol{\theta}_2)\| \le L_{\boldsymbol{\theta}}\|\boldsymbol{\theta}_1 - \boldsymbol{\theta}_2\| \tag{2}$$

$$\forall \boldsymbol{\theta}, \boldsymbol{x}_1, \boldsymbol{x}_2, \ \|f_i(\boldsymbol{x}_1, \boldsymbol{\theta}) - f_i(\boldsymbol{x}_2, \boldsymbol{\theta})\| \le L_{\boldsymbol{x}}\|\boldsymbol{x}_1 - \boldsymbol{x}_2\| \tag{3}$$

We then study the first-order smoothness of the adversarial loss objective function $\mathcal{L}_\epsilon(\boldsymbol{x}, \boldsymbol{\theta})$.

**Theorem 4.2.** *(Lipschitz continuity of adversarial loss) If Assumption 4.1 holds, we have:*

$$\forall \boldsymbol{x}, \boldsymbol{\theta}_1, \boldsymbol{\theta}_2, \ \|\mathcal{L}_\epsilon(\boldsymbol{x}, \boldsymbol{\theta}_1) - \mathcal{L}_\epsilon(\boldsymbol{x}, \boldsymbol{\theta}_2)\| \le A_{\boldsymbol{\theta}}\|\boldsymbol{\theta}_1 - \boldsymbol{\theta}_2\|, \tag{4}$$

*The constant $A_{\boldsymbol{\theta}} = 2\sum_{i \in \mathcal{S}_+} y_i L_{\boldsymbol{\theta}}$ where $\mathcal{S}_+ = \{i \mid y_i \ge 0, h_i(\boldsymbol{x} + \boldsymbol{\delta}_1, \boldsymbol{\theta}_2) > h_i(\boldsymbol{x} + \boldsymbol{\delta}_1, \boldsymbol{\theta}_1)\}$, $\boldsymbol{\delta}_1 \in \arg\max_{\boldsymbol{\delta} \in \mathcal{S}_\epsilon} \mathcal{L}(\boldsymbol{x} + \boldsymbol{\delta}, \boldsymbol{\theta})$ and $\boldsymbol{\delta}_2 \in \arg\max_{\boldsymbol{\delta} \in \mathcal{S}_\epsilon} \mathcal{L}(\boldsymbol{x} + \boldsymbol{\delta}, \boldsymbol{\theta})$.*

The proof is deferred to Appendix B.1, in which we can see the upper bound in Theorem 4.2 is tight. Theorem 4.2 indicates that the adversarial loss $\mathcal{L}_\epsilon(\boldsymbol{x}, \boldsymbol{\theta})$ is Lipschitz continuous, consistent with [48].

To study the second-order smoothness of $\mathcal{L}_\epsilon(\boldsymbol{x}, \boldsymbol{\theta})$, we start with the following assumption.

**Assumption 4.3. (Second-order Lipschitz condition)** $\forall i \in \{0, 1, ..., K-1\}$, the function $f_i$ satisfies the following second-order Lipschitz conditions, with constants $L_{\boldsymbol{\theta\theta}}, L_{\boldsymbol{\theta x}}$:

$$\forall \boldsymbol{x}, \boldsymbol{\theta}_1, \boldsymbol{\theta}_2, \ \|\nabla_{\boldsymbol{\theta}} f_i(\boldsymbol{x}, \boldsymbol{\theta}_1) - \nabla_{\boldsymbol{\theta}} f_i(\boldsymbol{x}, \boldsymbol{\theta}_2)\| \le L_{\boldsymbol{\theta\theta}}\|\boldsymbol{\theta}_1 - \boldsymbol{\theta}_2\| \tag{5}$$

$$\forall \boldsymbol{\theta}, \boldsymbol{x}_1, \boldsymbol{x}_2, \ \|\nabla_{\boldsymbol{\theta}} f_i(\boldsymbol{x}_1, \boldsymbol{\theta}) - \nabla_{\boldsymbol{\theta}} f_i(\boldsymbol{x}_2, \boldsymbol{\theta})\| \le L_{\boldsymbol{\theta x}}\|\boldsymbol{x}_1 - \boldsymbol{x}_2\| \tag{6}$$

**Theorem 4.4.** *(Lipschitz smoothness of adversarial loss) If Assumption 4.1 and 4.3 hold, we have:*

$$\forall \boldsymbol{x}, \boldsymbol{\theta}_1, \boldsymbol{\theta}_2, \ \|\nabla_{\boldsymbol{\theta}} \mathcal{L}_\epsilon(\boldsymbol{x}, \boldsymbol{\theta}_1) - \nabla_{\boldsymbol{\theta}} \mathcal{L}_\epsilon(\boldsymbol{x}, \boldsymbol{\theta}_2)\| \le A_{\boldsymbol{\theta\theta}}\|\boldsymbol{\theta}_1 - \boldsymbol{\theta}_2\| + B_{\boldsymbol{\theta\delta}} \tag{7}$$

*The constants $A_{\boldsymbol{\theta\theta}} = L_{\boldsymbol{\theta\theta}}$ and $B_{\boldsymbol{\theta\delta}} = L_{\boldsymbol{\theta x}}\|\boldsymbol{\delta}_1 - \boldsymbol{\delta}_2\| + 4L_{\boldsymbol{\theta}}$ where $\boldsymbol{\delta}_1 \in \arg\max_{\boldsymbol{\delta} \in \mathcal{S}_\epsilon} \mathcal{L}(\boldsymbol{x} + \boldsymbol{\delta}, \boldsymbol{\theta}_1)$ and $\boldsymbol{\delta}_2 \in \arg\max_{\boldsymbol{\delta} \in \mathcal{S}_\epsilon} \mathcal{L}(\boldsymbol{x} + \boldsymbol{\delta}, \boldsymbol{\theta}_2)$.*

The proof is deferred to Appendix B.2. Theorem 4.4 indicates the adversarial loss objective function $\mathcal{L}_\epsilon(\boldsymbol{x}, \boldsymbol{\theta})$ w.r.t. the model parameter $\boldsymbol{\theta}$ is no longer smooth. That is to say, gradients in arbitrarily small neighborhoods in the $\boldsymbol{\theta}$-space can change discontinuously. Furthermore, the degree of discontinuity is indicated by the value of $B_{\boldsymbol{\theta\delta}} = L_{\boldsymbol{\theta x}}\|\boldsymbol{\delta}_1 - \boldsymbol{\delta}_2\| + 4L_{\boldsymbol{\theta}}$, so a larger $\|\boldsymbol{\delta}_1 - \boldsymbol{\delta}_2\|$ can intensify the gradient discontinuity. Additionally, as elucidated by *Theorem 2* in [48], the gradients are non-vanishing in adversarial training. A large $B_{\boldsymbol{\theta\delta}}$ introduces large gradient magnitudes asymptotically, making optimization challenging.

In practice, we may use non-smooth activations, like ReLU, which do not strictly satisfy Assumption 4.3. For example, the gradient of ReLU changes abruptly in the neighborhood around $0$. In this regard, we provide a more detailed analysis of this case in Appendix D, which suggests that our analyses can be straightforwardly extended to networks with non-smooth activations.

Without the loss of generality, the Lipschitz properties in Assumption 4.1 and 4.3 can be based on any *proper $l_p$ norm*, i.e., $p \in [1, +\infty]$, which, however, does not include $l_0$ norm. Correspondingly, $\|\boldsymbol{\delta}_1 - \boldsymbol{\delta}_2\|$ in the expression of $B_{\boldsymbol{\theta\delta}}$ is based on the same norm as in the assumptions. On the popular benchmark CIFAR-10, the commonly used values of $\epsilon$ in the $l_0$, $l_1$, $l_2$ and $l_\infty$ cases are $360^2$, 24, 0.5 and $8/255$, respectively [5, 41, 19, 23]. In Appendix E, we discuss the numerical upper bound of $\|\boldsymbol{\delta}_1 - \boldsymbol{\delta}_2\|$ when the Lipschitz assumptions are based on different proper norms. **The results demonstrate that the upper bound of $\|\boldsymbol{\delta}_1 - \boldsymbol{\delta}_2\|$ in the $l_0$ case is always significantly larger than other cases, indicating a more craggy adversarial loss function in $l_0$ adversarial training.** Moreover, to corroborate the Lipschitz smoothness assumption in Inequality (6), we compare the distances between the gradients induced by one-step and multi-step attacks in Appendix F.2.

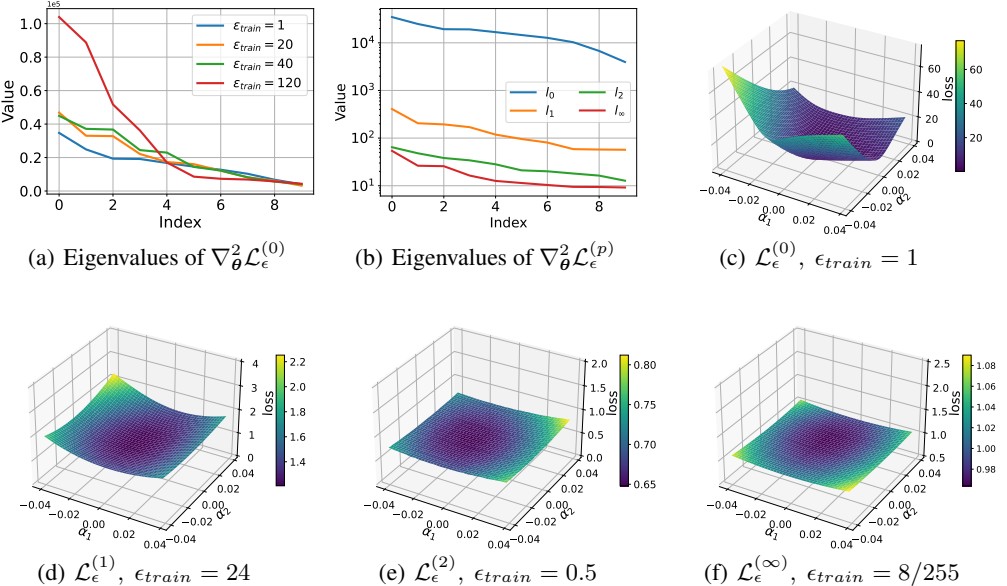

(a) Eigenvalues of $\nabla_{\boldsymbol{\theta}}^2 \mathcal{L}_\epsilon^{(0)}$   (b) Eigenvalues of $\nabla_{\boldsymbol{\theta}}^2 \mathcal{L}_\epsilon^{(p)}$   (c) $\mathcal{L}_\epsilon^{(0)}$, $\epsilon_{train} = 1$

(d) $\mathcal{L}_\epsilon^{(1)}$, $\epsilon_{train} = 24$   (e) $\mathcal{L}_\epsilon^{(2)}$, $\epsilon_{train} = 0.5$   (f) $\mathcal{L}_\epsilon^{(\infty)}$, $\epsilon_{train} = 8/255$

Figure 3: Smoothness of adversarial loss objective functions. All losses are calculated on the training set of CIFAR-10 by PreactResNet-18. The $l_0$, $l_1$, $l_2$ and $l_\infty$ models are obtained by 1-step sAT [23], Fast-EG-$l_1$ [19], 1-step PGD [38] and GradAlign [37], respectively. **(a)** Top 10 eigenvalues of $\nabla_{\boldsymbol{\theta}}^2 \mathcal{L}_\epsilon^{(0)}(\boldsymbol{x}, \boldsymbol{\theta})$ with different values of $\epsilon_{train}$ in the $l_0$ case. **(b)** Top 10 eigenvalues of $\nabla_{\boldsymbol{\theta}}^2 \mathcal{L}_\epsilon^{(p)}(\boldsymbol{x}, \boldsymbol{\theta})$ under different choices of $p$, including 0, 1, 2 and $\infty$. The y-axis is shown on a log scale. **(c) - (f)** The loss landscape of $\mathcal{L}_\epsilon(\boldsymbol{x}, \boldsymbol{\theta} + \alpha_1 \boldsymbol{v}_1 + \alpha_2 \boldsymbol{v}_2)$ where $\boldsymbol{v}_1$ and $\boldsymbol{v}_2$ are the eigenvectors associated with the top 2 eigenvalues of $\nabla_{\boldsymbol{\theta}}^2 \mathcal{L}_\epsilon(\boldsymbol{x}, \boldsymbol{\theta})$, respectively. **(c)** $l_0$ case, $\epsilon_{train} = 1$. **(d)** $l_1$ case, $\epsilon_{train} = 24$. **(e)** $l_2$ case, $\epsilon_{train} = 0.5$. **(f)** $l_\infty$ case, $\epsilon_{train} = 8/255$.

### 4.2 Numerical Validation

To validate the conclusions in theoretical analyses, we conduct numerical experiments to study the properties of loss landscape of $l_0$ adversarial training and compare it with the $l_\infty$, $l_2$ and $l_1$ cases.

We first study the curvature in the neighborhood of model parameters, which reflects the second-order smoothness of the loss function and is dominated by top eigenvalues of Hessian matrix $\nabla_{\boldsymbol{\theta}}^2 \mathcal{L}_\epsilon(\boldsymbol{x}, \boldsymbol{\theta})$. Numerically, we employ the power method [49, 48, 50] to iteratively estimate the eigenvalues and the corresponding eigenvectors of Hessian matrices. We plot the top-10 eigenvalues of the Hessian matrices $\nabla_{\boldsymbol{\theta}}^2 \mathcal{L}_\epsilon(\boldsymbol{x}, \boldsymbol{\theta})$ under different $\epsilon$ in $l_0$ cases in Figure 3 (a). In addition, we compare the Hessian spectrum in the $l_0$ case with $l_\infty$, $l_2$ and $l_1$ cases in Figure 3 (b). Our results in Figure 3 (a) demonstrate that eigenvalues of Hessian matrices in $l_0$ cases increase as $\epsilon$ grows, indicating a higher degree of non-smoothness for a larger $\epsilon$. Moreover, Figure 3 (b) indicates that the adversarial loss landscape in the $l_0$ case is more craggy than its $l_\infty$, $l_2$ and $l_1$ counterparts, even when we set $\epsilon = 1$, i.e., perturbing only a single pixel. These observations corroborate that $l_0$ adversarial training exhibits worse second-order smoothness than other cases, causing challenges in optimization.

To study the first-order smoothness, we visualize the loss landscape of different settings in Figures 3 (c)-(f), which demonstrate that the loss in the $l_0$ case abruptly increases even with subtle changes in the model parameters. This further suggests the non-smooth nature of the $l_0$ adversarial loss landscape. More loss landscape visualizations of $l_0$ adversarial training with different $\epsilon$ are provided in Appendix F.9. The observations are consistent with that in Figure 3. Accordingly, we confirm that the loss landscape of $l_0$ adversarial loss function is more craggy than other cases from both theoretical and empirical perspectives. In addition, among the cases studied in Figure 3, the $l_0$ cases are the only ones suffering from CO.

---

[2]In [23], the $l_0$ adversarial budget for training on CIFAR-10 is 120 *in the pixel space* of RGB images, so the equivalent $l_0$ norm *in the feature space* is 360. Note that the adversarial budget mentioned in the experiment part is still in the pixel space.

On the other side, we show in Figure 4 of Appendix F.1 that successful attempts to obtain robust models against $l_0$ bounded perturbation also include elements that help improve the smoothness of the loss landscape. 20-step sAT in [23] uses an early stopping (ES) strategy to avoid CO and to achieve competitive performance. Specifically, ES interrupts the attack iteration once the current perturbed input is misclassified. ES is shown to circumvent the potential for excessive gradient magnitude while maintaining the efficacy of the generated perturbations. This finding further highlights a strong correlation between CO and the craggy nature of the loss landscape in $l_0$ adversarial training.

In summary, our results suggest that **the $l_0$ adversarial training exhibits a more craggy loss landscape than other cases, which shows a strong correlation with CO**. Additionally, despite the non-trivial performance of 20-step sAT with ES, its performance still exhibits considerable fluctuation and can be further improved, underscoring the need for a smoother loss function. In the next subsection, we will offer our recipe to address the CO issue in fast $l_0$ adversarial training.

### 4.3 Recipe: Soft Label and Trade-off Loss Function Smooth Loss Landscape

Notice that $A_{\boldsymbol{\theta}}$ in Theorem 4.2 can be regarded as a function of the label $\boldsymbol{y}$. Let $\boldsymbol{y}_h \in \{0, 1\}^K$ and $\boldsymbol{y}_s \in (0, 1)^K$ denote the hard and soft label, respectively. That is to say, $\boldsymbol{y}_h$ is a one-hot vector, while $\boldsymbol{y}_s$ is a dense vector in a simplex. We find that **soft label $\boldsymbol{y}_s$ leads to a reduced first-order Lipschitz constant**, thereby enhancing the Lipschitz continuity of the adversarial loss function. Detailed theoretical analysis is deferred to Proposition C.1 of Appendix C.

However, as indicated by Theorem 4.4, the second-order Lipschitz constant remains unaffected by variations in $\boldsymbol{y}$. Considering the poor performance on clean inputs when CO happens, we introduce a trade-off loss function $\mathcal{L}_{\epsilon,\alpha}$ which interpolates between clean and adversarial losses: $\mathcal{L}_{\epsilon,\alpha}(\boldsymbol{x}, \boldsymbol{\theta}) = (1-\alpha)\mathcal{L}(\boldsymbol{x}, \boldsymbol{\theta}) + \alpha \max_{\boldsymbol{\delta} \in \mathcal{S}_\epsilon(\boldsymbol{x})} \mathcal{L}(\boldsymbol{x}+\boldsymbol{\delta}, \boldsymbol{\theta})$ where $\alpha \in [0, 1]$ is the interpolation factor. We find that **trade-off loss function can enhance the second-order smoothness of adversarial loss function**. Detailed theoretical analysis is deferred to Proposition C.2 of Appendix C.

In implementation, we can stabilize and improve the performance of fast adversarial training against $l_0$ bounded perturbations by combining soft labels and trade-off loss function. In addition, several available techniques, such as self-adaptive training (SAT) [40] and TRADES [30], can be considered variations of soft labels and trade-off loss function. In Appendix A, we provide the pseudo-codes of both SAT and TRADES and the formulation of their combination as a reference. It should be highlighted that the rationale for using soft labels and trade-off loss function is different for the $l_0$ case. Although they are widely leveraged to address robust overfitting to boost performance in $l_2$ and $l_\infty$ cases, **smoothing the loss function is essential to address the CO issue in the $l_0$ case**.

## 5 Experiments

In this section, we perform extensive experiments to investigate various approaches that can stabilize and improve the performance of fast $l_0$ adversarial training. Then, we compare the performance of 1-step adversarial training with the multi-step counterparts.

### 5.1 Approaches to Improving 1-Step $l_0$ Adversarial Training

Table 4: Comparison of different approaches and their combinations in robust accuracy (%) by sAA. The target sparsity level $\epsilon = 20$. We compare PreAct ResNet-18 [24] models trained on CIFAR-10 [25]. Note that **S** and **N** denote SAT and N-FGSM, respectively. The *italic numbers* indicate catastrophic overfitting (CO) happens.

| Method | sAT | Trade-off | sTRADES (T) | sTRADES (F) |
|--------|-----|-----------|-------------|-------------|
| 1-step | *0.0* | *2.6* | 31.0 | 55.4 |
| + N | *0.3* | *17.5* | 46.9 | 55.9 |
| + S | 29.3 | 30.3 | 61.4 | 59.4 |
| + S&N | 43.8 | 39.2 | **63.0** | 62.6 |

We begin our analysis by evaluating the effectiveness of different approaches and their combinations, focusing on those that incorporate either soft labels or trade-off loss functions. Additionally, we explore a data augmentation technique N-FGSM [43], which may mitigate the sub-optimality of

Table 5: Robust accuracy (%) against sparse attacks. **(a)** PreActResNet-18 trained on **CIFAR-10**, where the attack sparsity level $\epsilon = 20$. **(b)** ResNet-34 trained on **ImageNet-100**, where $\epsilon = 200$. CornerSearch (CS) is not evaluated due to its high computational complexity. Cost times are recorded on one NVIDIA RTX 6000 Ada.

(a) **CIFAR-10,** $\epsilon = 20$

| Model | Time Cost | Clean | Black | | White | | | | sAA |
|---|---|---|---|---|---|---|---|---|---|
| | | | CS | RS | SAIF | $\sigma$-zero | sPGD$_p$ | sPGD$_u$ | |
| *Multi-step* | | | | | | | | | |
| sAT | 5.3 h | 84.5 | 52.1 | 36.2 | 76.6 | 79.8 | 75.9 | 75.3 | 36.2 |
| **+S&N** | 5.5 h | 80.8 | 64.1 | 61.1 | 76.1 | 78.7 | 76.8 | 75.1 | 61.0 |
| sTRADES | 5.5 h | 89.8 | 69.9 | 61.8 | 84.9 | 85.9 | 84.6 | 81.7 | 61.7 |
| **+S&N** | 5.4 h | 82.2 | 66.3 | 66.1 | 77.1 | 77.0 | 74.1 | 72.2 | **65.5** |
| *One-step* | | | | | | | | | |
| **Fast-LS-**$l_0$ | 0.8 h | 82.5 | 69.3 | 65.4 | 75.7 | 73.7 | 67.2 | 67.7 | **63.0** |

(b) **ImageNet-100,** $\epsilon = 200$

| Model | Time Cost | Clean | Black | White | | | | sAA |
|---|---|---|---|---|---|---|---|---|
| | | | RS | SAIF | $\sigma$-zero | sPGD$_p$ | sPGD$_u$ | |
| *Multi-step* | | | | | | | | |
| sAT | 325 h | 86.2 | 61.4 | 69.0 | 78.6 | 78.0 | 77.8 | 61.2 |
| **+S&N** | 336 h | 83.0 | 75.0 | 76.4 | 80.8 | 78.8 | 79.2 | 74.8 |
| sTRADES | 359 h | 84.8 | 76.0 | 77.4 | 81.6 | 80.6 | 81.4 | 75.8 |
| **+S&N** | 360 h | 82.4 | 78.2 | 79.2 | 80.0 | 78.2 | 79.8 | **77.8** |
| *One-step* | | | | | | | | |
| **Fast-LS-**$l_0$ | 44 h | 82.4 | 76.8 | 75.4 | 74.0 | 74.6 | 74.6 | **72.4** |

perturbation location by randomly perturbing more pixels. Our findings, summarized in Table 4, are all based on 1-step adversarial training. The robust accuracy is measured using the sparse-AutoAttack (sAA) method, with $\epsilon$ set to 20.

In Table 4, we investigate the following approaches and their combinations: **(1) sAT**: adversarial training against 1-step sPGD [23]. **(2) Tradeoff**: 1-step adversarial training with the trade-off loss function as discussed in Section 4.3. **(3) sTRADES**: the 1-step sTRADES [23]. As discussed in Appendix A, it incorporates both soft label and trade-off loss function. We include two variants of sTRADES for comparison: **sTRADES (T)** is the training mode where we only use the loss objective function of TRADES for training but still use the cross-entropy loss to generate adversarial examples; **sTRADES (F)** is the full mode where we use the KL divergence loss function for generating adversarial perturbations. Compared with 1-step sAT, sTRADES (T) introduces $25\%$ overhead while sTRADES (F) introduces $50\%$ overhead. **(4) SAT (S)**: self-adaptive training [40]. As discussed in Appendix A, it introduces soft labels based on the moving average of the historical predictions and uses adaptive weights for training instances of different prediction confidence. **(5) N-FGSM (N)**: data augmentation technique by adding random noise to the training data. It is proven effective in 1-step adversarial training [43]. Among these approaches, SAT (S) and sTRADES introduce soft labels; trade-off and sTRADES use trade-off loss functions to smooth the loss objective function. The comparison with other smoothing approaches is undertaken in Appendix F.3. The implementation details are deferred to Appendix G.

The results in Table 4 indicate that using trade-off loss function alone still suffers from CO. In contrast, combining it with soft label, either by SAT or sTRADES, can eliminate CO and achieve notable robust accuracy. Furthermore, sTRADES (F) outperforms sTRADES (T) by a substantial margin of $24.4\%$, which can be attributed to the generation of higher-quality adversarial examples for training by sTRADES (F). Finally, both SAT and N-FGSM can enhance the performance of all approaches, demonstrating their effectiveness.

Note that the results presented in Table 4 are obtained using sAA, which is known for generating the strongest attacks in terms of $l_0$ perturbations. Our findings demonstrate that incorporating soft labels and trade-off loss function yields substantial performance improvements in 1-step $l_0$ adversarial training. Among various combinations of methods explored, the model trained with sTRADES (T) with SAT and N-FGSM achieves the highest robust accuracy against sAA, reaching an impressive $63.0\%$. This establishes the state-of-the-art performance in fast $l_0$ adversarial training methods. For convenience, we name this combination (i.e., 1-step sTRADES + SAT + N-FGSM) **Fast-L**oss **S**moothing-$l_0$ (**Fast-LS-**$l_0$). Its pseudo-code is given in Algorithm 3 of Appendix A.

Furthermore, we validate the robustness of our method against feature-space attacks in Appendix F.6, validate the efficacy of our method for $l_1$, $l_2$ and $l_\infty$ cases in Appendix F.7, validate the efficacy of our method on different networks in Appendix F.8, visualize the loss landscape when using our method in Appendix F.10 to demonstrate its improved smoothness, and conduct ablation studies for analysis in Appendix F.11. Our extensive results demonstrate the broad effectiveness of smoothing loss landscape in fast adversarial training. Nevertheless, it turns out essential to address CO and has the largest performance improvement in the $l_0$ cases compared with other $l_p$ norms ($p \geq 1$), because the loss landscape is much more craggy in the $l_0$ cases, as analyzed in Section 4.

## 5.2 Comparison with Multi-Step Adversarial Training

In this section, we compare 1-step adversarial training with its multi-step counterpart. For multi-step adversarial training, we follow the settings in [23] and use 20-step sPGD based on cross-entropy to generate adversarial perturbations in sAT and sTRADES. Similar to Table 4, we incorporate SAT and N-FGSM into multi-step adversarial training as well. For 1-step adversarial training, we focus on the best configuration in Table 4, i.e., Fast-LS-$l_0$.

We conduct extensive experiments on various datasets. The results on CIFAR-10 and ImageNet-100 [51] are demonstrated in Table 5. More results on CIFAR-100 [25] and GTSRB [52] are in Table 8 and 9 of Appendix F.4, respectively. In addition to the performance under sAA, we report the robust accuracy under various black-box and white box attacks, including CornerSearch (CS) [21], Sparse-RS (RS) [22], SAIF [42], $\sigma$-zero[3] [53] and two versions of sPGD [23]. Moreover, we report the clean accuracy and running time for reference. Finally, to more comprehensively validate the effectiveness of our results, we report the standard deviation of the performance in Table 10 of Appendix F.5.

The results in Table 5, 8 and 9 suggest that our method can improve the performance of both 1-step and multi-step adversarial training. In addition, N-FGSM, originally designed for 1-step adversarial training, also contributes to performance improvements in the multi-step scenario. Furthermore, these techniques can greatly narrow down the performance gaps between 1-step and multi-step adversarial training, making fast adversarial training more feasible and competitive. With the assistance of SAT and N-FGSM, **our Fast-LS-$l_0$ can achieve a performance that is merely $2.5\%$ lower than that of the $20$-step sTRADES while requiring less than $1/6$ of the total running time**.

## 6  Conclusion

We highlight that the catastrophic overfitting (CO) in $l_0$ adversarial training is caused by sub-optimal perturbation locations. Our analyses show that the loss landscape of $l_0$ adversarial training is more craggy than other cases and correlates with CO. To address this, we utilize soft labels and a trade-off loss function to smooth the adversarial loss. Extensive experiments show our method effectively mitigates CO and narrows the performance gap between 1-step and multi-step $l_0$ adversarial training, achieving state-of-the-art robustness against sparse attacks.

## Broader Impacts and Limitations

Since our method is evaluated on benchmarks, we do not see it has an obvious negative societal impact. Despite the effectiveness of soft label and trade-off loss function, we cannot guarantee they are the optimal method to smooth the loss landscape of $l_0$ adversarial training. We leave developing more effective methods as future work.

## Acknowledgments and Disclosure of Funding

This work is supported by National Natural Science Foundation of China (NSFC Project No. 62306250) and City University of Hong Kong (CityU Project No. 9610614).

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

# A   Algorithm Details

---

**Algorithm 1** Self-Adaptive Training (SAT) [40]

---

1: **Input:** Data: $\{(\boldsymbol{x}_i, \boldsymbol{y}_i)\}_n$; Initial target $\{\boldsymbol{t}_i\}_n = \{\boldsymbol{y}_i\}_n$; Batch size: $m$; Classifier: $f$; Enabling epoch: $E_s$; Momentum factor: $\alpha$
2: **repeat**
3:     Fetch mini-batch data $\{(\boldsymbol{x}_i, \boldsymbol{t}_i)\}_m$ at current epoch $e$
4:     **for** $i = 1, ..., m$ **do**
5:       $\boldsymbol{p}_i = \mathrm{softmax}(f(\boldsymbol{x}_i))$
6:       **if** $e > E_s$ **then**
7:         $\boldsymbol{t}_i = \alpha \times \boldsymbol{t}_i + (1 - \alpha) \times \boldsymbol{p}_i$
8:       **end if**
9:       $w_i = \max_j \boldsymbol{t}_{i,j}$
10:    **end for**
11:    Calculate the loss $\mathcal{L}_{SAT} = -\frac{1}{\sum_i w_i} \sum_i w_i \sum_j \boldsymbol{t}_{i,j} \log \boldsymbol{p}_{i,j}$
12:    Update the parameters of $f$ on $\mathcal{L}_{SAT}$
13: **until** end of training

---

---

**Algorithm 2** TRADES [30]

---

1: **Input:** Data: $(\boldsymbol{x}, \boldsymbol{y})$; Classifier: $f$; Balancing factor: $\beta$; TRADES mode: $mode$; Sparse level: $\epsilon$
2: **if** $mode = F$ **then**
3:    Generate adversarial sample $\widetilde{\boldsymbol{x}} = \max_{(\widetilde{\boldsymbol{x}}-\boldsymbol{x}) \in \mathcal{S}_\epsilon(\boldsymbol{x})} \mathrm{KL}(f(\boldsymbol{x}), f(\widetilde{\boldsymbol{x}}))$
4: **else if** $mode = T$ **then**
5:    Generate adversarial sample $\widetilde{\boldsymbol{x}} = \max_{(\widetilde{\boldsymbol{x}}-\boldsymbol{x}) \in \mathcal{S}_\epsilon(\boldsymbol{x})} \mathrm{CE}(f(\widetilde{\boldsymbol{x}}), \boldsymbol{y})$
6: **end if**
7: Calculate the loss $\mathcal{L}_{TRADES} = \mathrm{CE}(f(\boldsymbol{x}), \boldsymbol{y}) + \beta \cdot \mathrm{KL}(f(\boldsymbol{x}), f(\widetilde{\boldsymbol{x}}))$
8: Update the parameters of $f$ on $\mathcal{L}_{TRADES}$

---

The pseudo-codes of SAT [40] and TRADES [30] are provided in Algorithm 1 and 2, respectively. For SAT, the moving average of the previous predictions $\{\boldsymbol{t}_i\}^n$ can be regarded as the soft labels. For TRADES, $f(\boldsymbol{x})$ can be seen as the soft label of $f(\widetilde{\boldsymbol{x}})$, and the combination of cross-entropy and KL divergence is also a trade-off loss function. Note that when combining SAT and TRADES, the loss $\mathcal{L}_{S+T}$ for a mini-batch data $\{(\boldsymbol{x}_i, \boldsymbol{y}_i)\}_m$ can be written as:

$$\mathcal{L}_{S+T} = -\frac{1}{\sum_i w_i} \sum_i w_i \cdot \mathrm{CE}(f(\boldsymbol{x}_i), \boldsymbol{t}_i) + \frac{\beta}{m} \sum_i \mathrm{KL}(f(\boldsymbol{x}_i), f(\widetilde{\boldsymbol{x}}_i)) \tag{8}$$

In addition, we provide the pseudo-code of the proposed Fast-LS-$l_0$, which incorporates SAT, TRADES and N-FGSM, in Algorithm 3.

# B   Proofs

## B.1   Proof of Theorem 4.2

*Proof.* Based on the definition of $\boldsymbol{\delta}_1$ and $\boldsymbol{\delta}_2$, we have $\mathcal{L}_\epsilon(\boldsymbol{x}, \boldsymbol{\theta}_1) = \mathcal{L}(\boldsymbol{x} + \boldsymbol{\delta}_1, \boldsymbol{\theta}_1)$ and $\mathcal{L}_\epsilon(\boldsymbol{x}, \boldsymbol{\theta}_2) = \mathcal{L}(\boldsymbol{x} + \boldsymbol{\delta}_2, \boldsymbol{\theta}_2)$. In this regard, we have:

$$\|\mathcal{L}_\epsilon(\boldsymbol{x}, \boldsymbol{\theta}_1) - \mathcal{L}_\epsilon(\boldsymbol{x}, \boldsymbol{\theta}_2)\| = \|\mathcal{L}(\boldsymbol{x} + \boldsymbol{\delta}_1, \boldsymbol{\theta}_1) - \mathcal{L}(\boldsymbol{x} + \boldsymbol{\delta}_2, \boldsymbol{\theta}_2)\| \tag{9}$$

When $\mathcal{L}(\boldsymbol{x} + \boldsymbol{\delta}_1, \boldsymbol{\theta}_1) \geq \mathcal{L}(\boldsymbol{x} + \boldsymbol{\delta}_2, \boldsymbol{\theta}_2)$ we have

$$
\begin{aligned}
&\|\mathcal{L}(\boldsymbol{x} + \boldsymbol{\delta}_1, \boldsymbol{\theta}_1) - \mathcal{L}(\boldsymbol{x} + \boldsymbol{\delta}_2, \boldsymbol{\theta}_2)\| \\
=&\|\mathcal{L}(\boldsymbol{x} + \boldsymbol{\delta}_1, \boldsymbol{\theta}_1) - \mathcal{L}(\boldsymbol{x} + \boldsymbol{\delta}_1, \boldsymbol{\theta}_2) + \mathcal{L}(\boldsymbol{x} + \boldsymbol{\delta}_1, \boldsymbol{\theta}_2) - \mathcal{L}(\boldsymbol{x} + \boldsymbol{\delta}_2, \boldsymbol{\theta}_2)\| \\
\leq&\|\mathcal{L}(\boldsymbol{x} + \boldsymbol{\delta}_1, \boldsymbol{\theta}_1) - \mathcal{L}(\boldsymbol{x} + \boldsymbol{\delta}_1, \boldsymbol{\theta}_2)\|
\end{aligned}
\tag{10}
$$

The inequality above is derived from the optimality of $\delta_2$, which indicates $\mathcal{L}(\boldsymbol{x} + \boldsymbol{\delta}_1, \boldsymbol{\theta}_2) - \mathcal{L}(\boldsymbol{x} + \boldsymbol{\delta}_2, \boldsymbol{\theta}_2) \leq 0$ and the assumption $\mathcal{L}(\boldsymbol{x} + \boldsymbol{\delta}_1, \boldsymbol{\theta}_1) \geq \mathcal{L}(\boldsymbol{x} + \boldsymbol{\delta}_2, \boldsymbol{\theta}_2)$.

**Algorithm 3** Fast-LS-$l_0$

---

1: **Input:** Data: $\{(\boldsymbol{x}_i, \boldsymbol{y}_i)\}^n$; Initial target $\{\boldsymbol{t}_i\}^n = \{\boldsymbol{y}_i\}^n$; Batch size: $m$; Classifier: $f$; Enabling epoch: $E_s$; Momentum factor: $\alpha$; Balancing factor: $\beta$; TRADES mode: $mode$; Sparse level: $\epsilon$
2: **repeat**
3:    Fetch mini-batch data $\{(\boldsymbol{x}_i, \boldsymbol{t}_i)\}_m$ at current epoch $e$
4:    **for** $i = 1, ..., m$ **do**
5:       $\boldsymbol{\eta}_i \sim \mathcal{S}_{2\epsilon}(\boldsymbol{x}_i)$
6:       $\boldsymbol{x}_i = \boldsymbol{x}_i + \boldsymbol{\eta}_i$             // *Augment sample with additive noise*
7:       **if** $mode = F$ **then**
8:          $\widetilde{\boldsymbol{x}}_i = \max_{(\widetilde{\boldsymbol{x}}_i - \boldsymbol{x}_i) \in \mathcal{S}_\epsilon(\boldsymbol{x}_i)} \text{KL}(f(\boldsymbol{x}_i), f(\widetilde{\boldsymbol{x}}_i))$
9:       **else if** $mode = T$ **then**
10:         $\widetilde{\boldsymbol{x}}_i = \max_{(\widetilde{\boldsymbol{x}}_i - \boldsymbol{x}_i) \in \mathcal{S}_\epsilon(\boldsymbol{x}_i)} \text{CE}(f(\widetilde{\boldsymbol{x}}_i), \boldsymbol{t}_i)$
11:       **end if**
12:       $\boldsymbol{p}_i = \text{softmax}(f(\boldsymbol{x}_i))$
13:       **if** $e > E_s$ **then**
14:          $\boldsymbol{t}_i = \alpha \times \boldsymbol{t}_i + (1 - \alpha) \times \boldsymbol{p}_i$
15:       **end if**
16:       $w_i = \max_j \boldsymbol{t}_{i,j}$
17:    **end for**
18:    Calculate $\mathcal{L}_{S+T}$ in Eq. (8)
19:    Update the parameters of $f$ on $\mathcal{L}_{S+T}$
20: **until** end of training

---

Similarly, when $\mathcal{L}(\boldsymbol{x} + \boldsymbol{\delta}_1, \boldsymbol{\theta}_1) \leq \mathcal{L}(\boldsymbol{x} + \boldsymbol{\delta}_2, \boldsymbol{\theta}_2)$ we have

$$
\begin{aligned}
&\|\mathcal{L}(\boldsymbol{x} + \boldsymbol{\delta}_1, \boldsymbol{\theta}_1) - \mathcal{L}(\boldsymbol{x} + \boldsymbol{\delta}_2, \boldsymbol{\theta}_2)\| \\
=&\|\mathcal{L}(\boldsymbol{x} + \boldsymbol{\delta}_1, \boldsymbol{\theta}_1) - \mathcal{L}(\boldsymbol{x} + \boldsymbol{\delta}_2, \boldsymbol{\theta}_1) + \mathcal{L}(\boldsymbol{x} + \boldsymbol{\delta}_2, \boldsymbol{\theta}_1) - \mathcal{L}(\boldsymbol{x} + \boldsymbol{\delta}_2, \boldsymbol{\theta}_2)\| \\
\leq&\|\mathcal{L}(\boldsymbol{x} + \boldsymbol{\delta}_2, \boldsymbol{\theta}_1) - \mathcal{L}(\boldsymbol{x} + \boldsymbol{\delta}_2, \boldsymbol{\theta}_2)\|
\end{aligned}
\tag{11}
$$

Without the loss of generality, we further bound $\|\mathcal{L}_\epsilon(\boldsymbol{x}, \boldsymbol{\theta}_1) - \mathcal{L}_\epsilon(\boldsymbol{x}, \boldsymbol{\theta}_2)\|$ based on (10). The derivation can be straightforwardly extended to (11) by replacing $\boldsymbol{\delta}_1$ with $\boldsymbol{\delta}_2$.

Based on the formulation of $\mathcal{L}$ in (1), $\|\mathcal{L}_\epsilon(\boldsymbol{x}, \boldsymbol{\theta}_1) - \mathcal{L}_\epsilon(\boldsymbol{x}, \boldsymbol{\theta}_2)\|$ can be further derived as follows:

$$
\begin{aligned}
\|\mathcal{L}_\epsilon(\boldsymbol{x}, \boldsymbol{\theta}_1) - \mathcal{L}_\epsilon(\boldsymbol{x}, \boldsymbol{\theta}_2)\| &\leq \left| \sum_{i \in \mathcal{S}_+} y_i \log \frac{h_i(\boldsymbol{x} + \boldsymbol{\delta}_1, \boldsymbol{\theta}_2)}{h_i(\boldsymbol{x} + \boldsymbol{\delta}_1, \boldsymbol{\theta}_1)} \right| \\
&= \sum_{i \in \mathcal{S}_+} y_i \left| \log \frac{1 + \sum_{j \neq i} \exp(f_j(\boldsymbol{x} + \boldsymbol{\delta}_1, \boldsymbol{\theta}_2) - f_i(\boldsymbol{x} + \boldsymbol{\delta}_1, \boldsymbol{\theta}_2))}{1 + \sum_{j \neq i} \exp(f_j(\boldsymbol{x} + \boldsymbol{\delta}_1, \boldsymbol{\theta}_1) - f_i(\boldsymbol{x} + \boldsymbol{\delta}_1, \boldsymbol{\theta}_1))} \right|
\end{aligned}
\tag{12}
$$

where $\mathcal{S}_+ = \{i \mid y_i \geq 0, h_i(\boldsymbol{x} + \boldsymbol{\delta}_1, \boldsymbol{\theta}_2) > h_i(\boldsymbol{x} + \boldsymbol{\delta}_1, \boldsymbol{\theta}_1)\}$. Then, according to the mediant inequality, we have

$$
\begin{aligned}
&\left| \log \frac{1 + \sum_{j \neq i} \exp(f_j(\boldsymbol{x} + \boldsymbol{\delta}_1, \boldsymbol{\theta}_2) - f_i(\boldsymbol{x} + \boldsymbol{\delta}_1, \boldsymbol{\theta}_2))}{1 + \sum_{j \neq i} \exp(f_j(\boldsymbol{x} + \boldsymbol{\delta}_1, \boldsymbol{\theta}_1) - f_i(\boldsymbol{x} + \boldsymbol{\delta}_1, \boldsymbol{\theta}_1))} \right| \\
\leq &\left| \log \frac{\sum_{j \neq i} \exp(f_j(\boldsymbol{x} + \boldsymbol{\delta}_1, \boldsymbol{\theta}_2) - f_i(\boldsymbol{x} + \boldsymbol{\delta}_1, \boldsymbol{\theta}_2))}{\sum_{j \neq i} \exp(f_j(\boldsymbol{x} + \boldsymbol{\delta}_1, \boldsymbol{\theta}_1) - f_i(\boldsymbol{x} + \boldsymbol{\delta}_1, \boldsymbol{\theta}_1))} \right| \\
\leq &\max_k \left| \log \frac{\exp(f_k(\boldsymbol{x} + \boldsymbol{\delta}_1, \boldsymbol{\theta}_2) - f_i(\boldsymbol{x} + \boldsymbol{\delta}_1, \boldsymbol{\theta}_2))}{\exp(f_k(\boldsymbol{x} + \boldsymbol{\delta}_1, \boldsymbol{\theta}_1) - f_i(\boldsymbol{x} + \boldsymbol{\delta}_1, \boldsymbol{\theta}_1))} \right| \\
\leq &\max_k |f_k(\boldsymbol{x} + \boldsymbol{\delta}_1, \boldsymbol{\theta}_2) - f_k(\boldsymbol{x} + \boldsymbol{\delta}_1, \boldsymbol{\theta}_1)| + |f_i(\boldsymbol{x} + \boldsymbol{\delta}_1, \boldsymbol{\theta}_2) - f_i(\boldsymbol{x} + \boldsymbol{\delta}_1, \boldsymbol{\theta}_1)| \\
\leq &2L_{\boldsymbol{\theta}} \|\boldsymbol{\theta}_1 - \boldsymbol{\theta}_2\|
\end{aligned}
\tag{13}
$$

Note that the bound on the right of (13) is tight. The upper bound can be achieved asymptotically if the condition in (14) and the Lipschitz bound in Assumption 4.1 are satisfied.

$$\left| |f_k(\boldsymbol{x} + \boldsymbol{\delta}_1, \boldsymbol{\theta}_2) - f_i(\boldsymbol{x} + \boldsymbol{\delta}_1, \boldsymbol{\theta}_2)| - |f_k(\boldsymbol{x} + \boldsymbol{\delta}_1, \boldsymbol{\theta}_1) - f_i(\boldsymbol{x} + \boldsymbol{\delta}_1, \boldsymbol{\theta}_1)| \right|$$
$$\gg \max_{j \neq k} \left| |f_j(\boldsymbol{x} + \boldsymbol{\delta}_1, \boldsymbol{\theta}_2) - f_i(\boldsymbol{x} + \boldsymbol{\delta}_1, \boldsymbol{\theta}_2)| - |f_j(\boldsymbol{x} + \boldsymbol{\delta}_1, \boldsymbol{\theta}_1) - f_i(\boldsymbol{x} + \boldsymbol{\delta}_1, \boldsymbol{\theta}_1)| \right| \tag{14}$$

Combining (9)-(13), we have

$$\|\mathcal{L}_\epsilon(\boldsymbol{x}, \boldsymbol{\theta}_1) - \mathcal{L}_\epsilon(\boldsymbol{x}, \boldsymbol{\theta}_2)\| \leq A_{\boldsymbol{\theta}} \|\boldsymbol{\theta}_1 - \boldsymbol{\theta}_2\|, \tag{15}$$

where $A_{\boldsymbol{\theta}} = 2 \sum_{i \in \mathcal{S}_+} y_i L_{\boldsymbol{\theta}}$. $\qquad\square$

## B.2 Proof of Theorem 4.4

*Proof.* Given (1), $\nabla_{\boldsymbol{\theta}} \mathcal{L}$ is computed as

$$\begin{aligned}
\nabla_{\boldsymbol{\theta}} \mathcal{L}(\boldsymbol{x}, \boldsymbol{\theta}) &= -\sum_{i=0}^{K-1} y_i \left[ \nabla_{\boldsymbol{\theta}} f_i(\boldsymbol{x}, \boldsymbol{\theta}) - \frac{\sum_j \exp(f_j(\boldsymbol{x}, \boldsymbol{\theta})) \nabla_{\boldsymbol{\theta}} f_j(\boldsymbol{x}, \boldsymbol{\theta})}{\sum_j \exp(f_j(\boldsymbol{x}, \boldsymbol{\theta}))} \right] \\
&= \frac{\sum_j \exp(f_j(\boldsymbol{x}, \boldsymbol{\theta})) \nabla_{\boldsymbol{\theta}} f_j(\boldsymbol{x}, \boldsymbol{\theta})}{\sum_j \exp(f_j(\boldsymbol{x}, \boldsymbol{\theta}))} - \sum_{i=0}^{K-1} y_i \nabla_{\boldsymbol{\theta}} f_i(\boldsymbol{x}, \boldsymbol{\theta}) \\
&\stackrel{\text{def}}{=} \sum_{j=0}^{K-1} h_j(\boldsymbol{x}, \boldsymbol{\theta}) \nabla_{\boldsymbol{\theta}} f_j(\boldsymbol{x}, \boldsymbol{\theta}) - \sum_{i=0}^{K-1} y_i \nabla_{\boldsymbol{\theta}} f_i(\boldsymbol{x}, \boldsymbol{\theta})
\end{aligned} \tag{16}$$

The second equality is based on the fact that $\{y_i\}_{i=0}^{K-1}$ is in a simplex. To simplify the notation, the last equation is based on the definition that $\{h_j\}_{j=0}^{K-1}$ is the result of softmax function applied to $\{f_j\}_{j=0}^{K-1}$, i.e., $h_j(\boldsymbol{x}, \boldsymbol{\theta}) = \frac{\exp(f_j(\boldsymbol{x}, \boldsymbol{\theta}))}{\sum_k \exp(f_k(\boldsymbol{x}, \boldsymbol{\theta}))}$. Therefore, we have $\sum_{j=0}^{K-1} h_j(\boldsymbol{x}, \boldsymbol{\theta}) = 1$ and $\forall j, h_j(\boldsymbol{x}, \boldsymbol{\theta}) > 0$.

According to the triangle inequality, we have:

$$\begin{aligned}
&\|\nabla_{\boldsymbol{\theta}_1} \mathcal{L}(\boldsymbol{x} + \boldsymbol{\delta}_1, \boldsymbol{\theta}_1) - \nabla_{\boldsymbol{\theta}_2} \mathcal{L}(\boldsymbol{x} + \boldsymbol{\delta}_2, \boldsymbol{\theta}_2)\| \\
&\leq \|\nabla_{\boldsymbol{\theta}_1} \mathcal{L}(\boldsymbol{x} + \boldsymbol{\delta}_1, \boldsymbol{\theta}_1) - \nabla_{\boldsymbol{\theta}_1} \mathcal{L}(\boldsymbol{x} + \boldsymbol{\delta}_2, \boldsymbol{\theta}_1)\| + \|\nabla_{\boldsymbol{\theta}_1} \mathcal{L}(\boldsymbol{x} + \boldsymbol{\delta}_2, \boldsymbol{\theta}_1) - \nabla_{\boldsymbol{\theta}_2} \mathcal{L}(\boldsymbol{x} + \boldsymbol{\delta}_2, \boldsymbol{\theta}_2)\|
\end{aligned} \tag{17}$$

Plug (16) to the first term on the right hand side of (17), we obtain:

$$\|\nabla_{\boldsymbol{\theta}_1} \mathcal{L}(\boldsymbol{x} + \boldsymbol{\delta}_1, \boldsymbol{\theta}_1) - \nabla_{\boldsymbol{\theta}_1} \mathcal{L}(\boldsymbol{x} + \boldsymbol{\delta}_2, \boldsymbol{\theta}_1)\| \leq \sum_{i=0}^{K-1} y_i \|\nabla_{\boldsymbol{\theta}_1} f_i(\boldsymbol{x} + \boldsymbol{\delta}_1, \boldsymbol{\theta}_1) - \nabla_{\boldsymbol{\theta}_1} f_i(\boldsymbol{x} + \boldsymbol{\delta}_2, \boldsymbol{\theta}_1)\|$$
$$+ \left\| \sum_{j=0}^{K-1} h_j(\boldsymbol{x} + \boldsymbol{\delta}_1, \boldsymbol{\theta}_1) \nabla_{\boldsymbol{\theta}} f_j(\boldsymbol{x} + \boldsymbol{\delta}_1, \boldsymbol{\theta}_1) - \sum_{j=0}^{K-1} h_j(\boldsymbol{x} + \boldsymbol{\delta}_2, \boldsymbol{\theta}_1) \nabla_{\boldsymbol{\theta}} f_j(\boldsymbol{x} + \boldsymbol{\delta}_2, \boldsymbol{\theta}_1) \right\| \tag{18}$$

The first term can be bounded based on Assumption 4.1. The second term can be bounded as follows:

$$\left\| \sum_{j=0}^{K-1} h_j(\boldsymbol{x} + \boldsymbol{\delta}_1, \boldsymbol{\theta}_1) \nabla_{\boldsymbol{\theta}} f_j(\boldsymbol{x} + \boldsymbol{\delta}_1, \boldsymbol{\theta}_1) - \sum_{j=0}^{K-1} h_j(\boldsymbol{x} + \boldsymbol{\delta}_2, \boldsymbol{\theta}_1) \nabla_{\boldsymbol{\theta}} f_j(\boldsymbol{x} + \boldsymbol{\delta}_2, \boldsymbol{\theta}_1) \right\|$$

$$\leq \left\| \sum_{j=0}^{K-1} h_j(\boldsymbol{x} + \boldsymbol{\delta}_1, \boldsymbol{\theta}_1) \nabla_{\boldsymbol{\theta}} f_j(\boldsymbol{x} + \boldsymbol{\delta}_1, \boldsymbol{\theta}_1) \right\| + \left\| \sum_{j=0}^{K-1} h_j(\boldsymbol{x} + \boldsymbol{\delta}_2, \boldsymbol{\theta}_1) \nabla_{\boldsymbol{\theta}} f_j(\boldsymbol{x} + \boldsymbol{\delta}_2, \boldsymbol{\theta}_1) \right\|$$

$$\leq \sum_{j=0}^{K-1} h_j(\boldsymbol{x} + \boldsymbol{\delta}_1, \boldsymbol{\theta}_1) \left\| \max_k \nabla_{\boldsymbol{\theta}} f_k(\boldsymbol{x} + \boldsymbol{\delta}_1, \boldsymbol{\theta}_1) \right\| + \sum_{j=0}^{K-1} h_j(\boldsymbol{x} + \boldsymbol{\delta}_2, \boldsymbol{\theta}_1) \left\| \max_k \nabla_{\boldsymbol{\theta}} f_k(\boldsymbol{x} + \boldsymbol{\delta}_2, \boldsymbol{\theta}_1) \right\|$$

$$\leq 2L_{\boldsymbol{\theta}} \tag{19}$$

Note that the bound on the right of (19) is tight. The first inequality is based on the triangle inequality. The second inequality and the third inequality can be achieved asymptotically when the equality of first-order Lipschitz continuity in Assumption 4.1 is achieved and the following condition is satisfied.

$$\exists k_1 \in \arg\max_i L_{\boldsymbol{\theta}}^{(i)}, h_{k_1}(\boldsymbol{x} + \boldsymbol{\delta}_1, \boldsymbol{\theta}_1) \to 1, \max_{j \neq k_1} h_j(\boldsymbol{x} + \boldsymbol{\delta}_1, \boldsymbol{\theta}_1) \to 0$$
$$\exists k_2 \in \arg\max_i L_{\boldsymbol{\theta}}^{(i)}, h_{k_2}(\boldsymbol{x} + \boldsymbol{\delta}_2, \boldsymbol{\theta}_1) \to 1, \max_{j \neq k_2} h_j(\boldsymbol{x} + \boldsymbol{\delta}_2, \boldsymbol{\theta}_1) \to 0 \tag{20}$$

Note that $k_1$ and $k_2$ are not always the same, since there may exist more than one biggest first-order Lipschitz constant.

Combining (18) and (19) together, we obtain:

$$\|\nabla_{\boldsymbol{\theta}_1} \mathcal{L}(\boldsymbol{x} + \boldsymbol{\delta}_1, \boldsymbol{\theta}_1) - \nabla_{\boldsymbol{\theta}_1} \mathcal{L}(\boldsymbol{x} + \boldsymbol{\delta}_2, \boldsymbol{\theta}_1)\| \leq 2L_{\boldsymbol{\theta}} + L_{\boldsymbol{\theta}\boldsymbol{x}} \|\boldsymbol{\delta}_2 - \boldsymbol{\delta}_1\| \tag{21}$$

Similarly, we have:

$$\|\nabla_{\boldsymbol{\theta}_1} \mathcal{L}(\boldsymbol{x} + \boldsymbol{\delta}_2, \boldsymbol{\theta}_1) - \nabla_{\boldsymbol{\theta}_2} \mathcal{L}(\boldsymbol{x} + \boldsymbol{\delta}_2, \boldsymbol{\theta}_2)\| \leq 2L_{\boldsymbol{\theta}} + L_{\boldsymbol{\theta}\boldsymbol{\theta}} \|\boldsymbol{\theta}_2 - \boldsymbol{\theta}_1\| \tag{22}$$

Combing the two inequalities above, we have:

$$\|\nabla_{\boldsymbol{\theta}} \mathcal{L}(\boldsymbol{x} + \boldsymbol{\delta}_1, \boldsymbol{\theta}_1) - \nabla_{\boldsymbol{\theta}} \mathcal{L}(\boldsymbol{x} + \boldsymbol{\delta}_2, \boldsymbol{\theta}_2)\| \leq A_{\boldsymbol{\theta}\boldsymbol{\theta}} \|\boldsymbol{\theta}_1 - \boldsymbol{\theta}_2\| + B_{\boldsymbol{\theta}\boldsymbol{\theta}} \tag{23}$$

where

$$A_{\boldsymbol{\theta}\boldsymbol{\theta}} = L_{\boldsymbol{\theta}\boldsymbol{\theta}}; \quad B_{\boldsymbol{\theta}\boldsymbol{\theta}} = 4L_{\boldsymbol{\theta}} + L_{\boldsymbol{\theta}\boldsymbol{x}} \|\boldsymbol{\delta}_1 - \boldsymbol{\delta}_2\| \tag{24}$$

$\square$

## C   Theoretical Analysis of Soft Labels and Trade-off Loss Function

We have the following proposition about soft labels:

**Proposition C.1.** *(Soft label improves Lipschitz continuity) Based on Theorem 4.2, given a hard label vector $\boldsymbol{y}_h \in \{0, 1\}^K$ and a soft label vector $\boldsymbol{y}_s \in (0, 1)^K$, we have $A_{\boldsymbol{\theta}}(\boldsymbol{y}_s) \leq A_{\boldsymbol{\theta}}(\boldsymbol{y}_h)$.*

*Proof.* For hard label $\boldsymbol{y}_h \in \{0, 1\}^K$, let that the $j$-th elements of $\boldsymbol{y}_h$ be 1 and the rest be 0. By the definition of $A_{\boldsymbol{\theta}}$ in Theorem 4.2, we have

$$A_{\boldsymbol{\theta}}(\boldsymbol{y}_h) = 2L_{\boldsymbol{\theta}}. \tag{25}$$

It is known that $\sum_{i=0}^{K-1} h_i(\boldsymbol{x}, \boldsymbol{\theta}) = 1$, which means $\exists j, h_j(\boldsymbol{x} + \boldsymbol{\delta}_1, \boldsymbol{\theta}_2) \leq h_j(\boldsymbol{x} + \boldsymbol{\delta}_1, \boldsymbol{\theta}_1)$. Then, for soft label $\boldsymbol{y}_s \in (0, 1)^K$, we have $|\mathcal{S}_+| < K$ where $\mathcal{S}_+ = \{i \mid y_i > 0, h_i(\boldsymbol{x} + \boldsymbol{\delta}_1, \boldsymbol{\theta}_2) > h_i(\boldsymbol{x} + \boldsymbol{\delta}_1, \boldsymbol{\theta}_1)\}$. Thus, it holds

$$A_{\boldsymbol{\theta}}(\boldsymbol{y}_s) = 2 \sum_{i \in \mathcal{S}_+} y_s^{(i)} L_{\boldsymbol{\theta}} \leq A_{\boldsymbol{\theta}}(\boldsymbol{y}_h). \tag{26}$$

The equality can be achieved asymptotically if $\sum_{i \notin \mathcal{S}_+} y_s^{(i)} \to 0$. $\square$

Proposition C.1 indicates that soft labels lead to a reduced first-order Lipschitz constant, thereby enhancing the Lipschitz continuity of the adversarial loss function.

Then, we have the following proposition about trade-off loss function:

**Proposition C.2.** *(Trade-off loss function improves Lipschitz smoothness) If Assumption 4.1 and 4.3 hold, we have:*

$$\|\nabla_\theta \mathcal{L}_{\epsilon,\alpha}(\boldsymbol{x}, \boldsymbol{\theta}_1) - \nabla_\theta \mathcal{L}_{\epsilon,\alpha}(\boldsymbol{x}, \boldsymbol{\theta}_2)\| \leq A_{\boldsymbol{\theta\theta}}\|\boldsymbol{\theta}_1 - \boldsymbol{\theta}_2\| + B'_{\boldsymbol{\theta\delta}} \tag{27}$$

*The constants* $A_{\boldsymbol{\theta\theta}} = L_{\boldsymbol{\theta\theta}}$ *and* $B'_{\boldsymbol{\theta\delta}} = \alpha L_{\boldsymbol{\theta x}}\|\boldsymbol{\delta}_1 - \boldsymbol{\delta}_2\| + 2(1 + \alpha)L_{\boldsymbol{\theta}}$ *where* $\boldsymbol{\delta}_1 \in \arg\max_{\boldsymbol{\delta} \in \mathcal{S}_\epsilon(\boldsymbol{x})} \mathcal{L}(\boldsymbol{x} + \boldsymbol{\delta}, \boldsymbol{\theta}_1)$ *and* $\boldsymbol{\delta}_2 \in \arg\max_{\boldsymbol{\delta} \in \mathcal{S}_\epsilon(\boldsymbol{x})} \mathcal{L}(\boldsymbol{x} + \boldsymbol{\delta}, \boldsymbol{\theta}_2)$.

*Proof.* By the definition of $\mathcal{L}_{\epsilon,\alpha}$ in Section 4.3, we have

$$\begin{aligned} &\|\nabla_{\boldsymbol{\theta}_1} \mathcal{L}_{\epsilon,\alpha}(\boldsymbol{x}, \boldsymbol{\theta}_1) - \nabla_{\boldsymbol{\theta}_2} \mathcal{L}_{\epsilon,\alpha}(\boldsymbol{x}, \boldsymbol{\theta}_2)\| \\ &\leq (1-\alpha)\|\nabla_{\boldsymbol{\theta}_1} \mathcal{L}(\boldsymbol{x}, \boldsymbol{\theta}_1) - \nabla_{\boldsymbol{\theta}_1} \mathcal{L}(\boldsymbol{x}, \boldsymbol{\theta}_2)\| + \alpha\|\nabla_{\boldsymbol{\theta}_1} \mathcal{L}_\epsilon(\boldsymbol{x}, \boldsymbol{\theta}_1) - \nabla_{\boldsymbol{\theta}_1} \mathcal{L}_\epsilon(\boldsymbol{x}, \boldsymbol{\theta}_2)\| \end{aligned} \tag{28}$$

According to (22) in the proof of Theorem 4.4, the first term of the right hand side of (28) can be derived as

$$\|\nabla_{\boldsymbol{\theta}_1} \mathcal{L}(\boldsymbol{x}, \boldsymbol{\theta}_1) - \nabla_{\boldsymbol{\theta}_2} \mathcal{L}(\boldsymbol{x}, \boldsymbol{\theta}_2)\| \leq L_{\boldsymbol{\theta\theta}}\|\boldsymbol{\theta}_1 - \boldsymbol{\theta}_2\| + 2L_{\boldsymbol{\theta}}. \tag{29}$$

According to Theorem 4.4, the second term of the right hand side of (28) satisifies

$$\|\nabla_{\boldsymbol{\theta}_1} \mathcal{L}_\epsilon(\boldsymbol{x}, \boldsymbol{\theta}_1) - \nabla_{\boldsymbol{\theta}_2} \mathcal{L}_\epsilon(\boldsymbol{x}, \boldsymbol{\theta}_2)\| \leq L_{\boldsymbol{\theta\theta}}\|\boldsymbol{\theta}_1 - \boldsymbol{\theta}_2\| + L_{\boldsymbol{\theta x}}\|\boldsymbol{\delta}_1 - \boldsymbol{\delta}_2\| + 4L_{\boldsymbol{\theta}}. \tag{30}$$

Combining (28), (29) and (30), we have

$$\|\nabla_{\boldsymbol{\theta}_1} \mathcal{L}_{\epsilon,\alpha}(\boldsymbol{x}, \boldsymbol{\theta}_1) - \nabla_{\boldsymbol{\theta}_2} \mathcal{L}_{\epsilon,\alpha}(\boldsymbol{x}, \boldsymbol{\theta}_2)\| \leq A_{\boldsymbol{\theta\theta}}\|\boldsymbol{\theta}_1 - \boldsymbol{\theta}_2\| + B'_{\boldsymbol{\theta\delta}}, \tag{31}$$

where $A_{\boldsymbol{\theta\theta}} = L_{\boldsymbol{\theta\theta}}$ and $B'_{\boldsymbol{\theta\delta}} = \alpha L_{\boldsymbol{\theta x}}\|\boldsymbol{\delta}_1 - \boldsymbol{\delta}_2\| + 2(1 + \alpha)L_{\boldsymbol{\theta}}$. □

According to Proposition C.2, the trade-off loss function $\mathcal{L}_{\epsilon,\alpha}$ enhances the second-order smoothness of adversarial loss objective function. Furthermore, compared with $l_1$, $l_2$ and $l_\infty$ cases, the trade-off loss function is particularly useful and necessary in the $l_0$ case. This is supported by the analyses in Section 4.1 and Appendix E, which demonstrate that $\|\boldsymbol{\delta}_1 - \boldsymbol{\delta}_2\|$ is much larger in $l_0$ bounded perturbations than other cases. Therefore, we expect the trade-off loss function $\mathcal{L}_{\epsilon,\alpha}$ can help mitigate CO by improving smoothness.

Similar to Theorem 4.4, Proposition C.2 can be straightforwardly extended to the networks with non-smooth activations, where Assumption 4.3 is not strictly satisfied. We provide a more detailed analysis in Appendix D to demonstrate the generality of our conclusions.

# D Theoretical Analysis of ReLU Networks

Similar to [48], we first make the following assumptions for the functions $\{f_i\}_{i=0}^{K-1}$ represented by a ReLU network.

**Assumption D.1.** $\forall i \in \{0, 1, ..., K-1\}$, the function $f_i$ satisfies the following conditions:

$$\forall \boldsymbol{x}, \boldsymbol{\theta}_1, \boldsymbol{\theta}_2, \quad \|f_i(\boldsymbol{x}, \boldsymbol{\theta}_1) - f_i(\boldsymbol{x}, \boldsymbol{\theta}_2)\| \leq L_{\boldsymbol{\theta}}\|\boldsymbol{\theta}_1 - \boldsymbol{\theta}_2\|, \tag{32}$$

$$\forall \boldsymbol{\theta}, \boldsymbol{x}_1, \boldsymbol{x}_2, \quad \|f_i(\boldsymbol{x}_1, \boldsymbol{\theta}) - f_i(\boldsymbol{x}_2, \boldsymbol{\theta})\| \leq L_{\boldsymbol{x}}\|\boldsymbol{x}_1 - \boldsymbol{x}_2\|, \tag{33}$$

$$\forall \boldsymbol{x}, \boldsymbol{\theta}_1, \boldsymbol{\theta}_2, \quad \|\nabla_{\boldsymbol{\theta}} f_i(\boldsymbol{x}, \boldsymbol{\theta}_1) - \nabla_{\boldsymbol{\theta}} f_i(\boldsymbol{x}, \boldsymbol{\theta}_2)\| \leq L_{\boldsymbol{\theta\theta}}\|\boldsymbol{\theta}_1 - \boldsymbol{\theta}_2\| + C_{\boldsymbol{\theta\theta}}, \tag{34}$$

$$\forall \boldsymbol{\theta}, \boldsymbol{x}_1, \boldsymbol{x}_2, \quad \|\nabla_{\boldsymbol{\theta}} f_i(\boldsymbol{x}_1, \boldsymbol{\theta}) - \nabla_{\boldsymbol{\theta}} f_i(\boldsymbol{x}_2, \boldsymbol{\theta})\| \leq L_{\boldsymbol{\theta x}}\|\boldsymbol{x}_1 - \boldsymbol{x}_2\| + C_{\boldsymbol{\theta x}}. \tag{35}$$

Compared to Assumption 4.1 and 4.3, we modify the the second-order smoothness assumptions by adding two constants $C_{\boldsymbol{\theta\theta}}$ and $C_{\boldsymbol{\theta x}}$, respectively. They denote the upper bound of the gradient difference in the neighborhood at non-smooth point. Thus, they quantify how drastically the (sub)gradients can change in a sufficiently small region in the parameter space.

Based on Assumption D.1, we have the following corollary:

**Corollary D.2.** *If Assumption D.1 is satisfied, it holds*

$$\|\mathcal{L}_\epsilon(\boldsymbol{x}, \boldsymbol{\theta}_1) - \mathcal{L}_\epsilon(\boldsymbol{x}, \boldsymbol{\theta}_2)\| \leq A_{\boldsymbol{\theta}} \|\boldsymbol{\theta}_1 - \boldsymbol{\theta}_2\|, \tag{36}$$

$$\|\nabla_{\boldsymbol{\theta}} \mathcal{L}_\epsilon(\boldsymbol{x}, \boldsymbol{\theta}_1) - \nabla_{\boldsymbol{\theta}} \mathcal{L}_\epsilon(\boldsymbol{x}, \boldsymbol{\theta}_2)\| \leq A_{\boldsymbol{\theta\theta}} \|\boldsymbol{\theta}_1 - \boldsymbol{\theta}_2\| + B_{\boldsymbol{\theta\delta}} + C_{\boldsymbol{\theta\theta}} + C_{\boldsymbol{\theta x}}. \tag{37}$$

*The Lipschitz constant* $A_{\boldsymbol{\theta}} = 2\sum_{i \in \mathcal{S}_+} y_i L_{\boldsymbol{\theta}}$, $A_{\boldsymbol{\theta\theta}} = L_{\boldsymbol{\theta\theta}}$ *and* $B_{\boldsymbol{\theta\delta}} = L_{\boldsymbol{\theta x}} \|\boldsymbol{\delta}_1 - \boldsymbol{\delta}_2\| + 4L_{\boldsymbol{\theta}}$ *where* $\boldsymbol{\delta}_1 \in \arg\max_{\boldsymbol{\delta} \in \mathcal{S}_\epsilon} \mathcal{L}(\boldsymbol{x} + \boldsymbol{\delta}, \boldsymbol{\theta}_1)$ *and* $\boldsymbol{\delta}_2 \in \arg\max_{\boldsymbol{\delta} \in \mathcal{S}_\epsilon} \mathcal{L}(\boldsymbol{x} + \boldsymbol{\delta}, \boldsymbol{\theta}_2)$.

The proof is similar to that of Theorem 4.2 and 4.4. Corollary D.2 indicates a more craggy loss landscape in the adversarial training of networks with non-smooth activations.

Additionally, the Proposition C.2 can be easily extended to accommodate Assumption D.1.

**Corollary D.3.** *If Assumption D.1 holds, then we have*

$$\|\nabla_{\boldsymbol{\theta}} \mathcal{L}_{\epsilon,\alpha}(\boldsymbol{x}, \boldsymbol{\theta}_1) - \nabla_{\boldsymbol{\theta}} \mathcal{L}_{\epsilon,\alpha}(\boldsymbol{x}, \boldsymbol{\theta}_2)\| \leq A_{\boldsymbol{\theta\theta}} \|\boldsymbol{\theta}_1 - \boldsymbol{\theta}_2\| + B'_{\boldsymbol{\theta\delta}} + C_{\boldsymbol{\theta\theta}} + C_{\boldsymbol{\theta x}}. \tag{38}$$

*The Lipschitz constant* $A_{\boldsymbol{\theta\theta}} = L_{\boldsymbol{\theta\theta}}$ *and* $B'_{\boldsymbol{\theta\delta}} = \alpha L_{\boldsymbol{\theta x}} \|\boldsymbol{\delta}_1 - \boldsymbol{\delta}_2\| + 2(1 + \alpha) L_{\boldsymbol{\theta}}$ *where* $\boldsymbol{\delta}_1 \in \arg\max_{\boldsymbol{\delta} \in \mathcal{S}_\epsilon} \mathcal{L}(\boldsymbol{x} + \boldsymbol{\delta}, \boldsymbol{\theta}_1)$ *and* $\boldsymbol{\delta}_2 \in \arg\max_{\boldsymbol{\delta} \in \mathcal{S}_\epsilon} \mathcal{L}(\boldsymbol{x} + \boldsymbol{\delta}, \boldsymbol{\theta}_2)$.

# E Discussion of the Upper Bound of $\|\boldsymbol{\delta}_1 - \boldsymbol{\delta}_2\|$

We define the $l_p$ adversarial budget for the perturbation $\boldsymbol{\delta} \in \mathbb{R}^d$ as $\mathcal{S}_\epsilon^{(p)} = \{\boldsymbol{\delta} \mid \|\boldsymbol{\delta}\|_p \leq \epsilon, \ 0 \leq \boldsymbol{x} + \boldsymbol{\delta} \leq 1\}$. Therefore, we have $\|\boldsymbol{\delta}_1 - \boldsymbol{\delta}_2\|_p \leq 2\epsilon$, and $\forall i, \ 0 \leq |\delta_1^{(i)} - \delta_2^{(i)}| \leq 1$ where $\delta_1^{(i)}$ and $\delta_2^{(i)}$ are the $i$-th element of $\boldsymbol{\delta}_1$ and $\boldsymbol{\delta}_2$, respectively. For convenience, we denote $\boldsymbol{\delta}_1 - \boldsymbol{\delta}_2$ as $\Delta\boldsymbol{\delta}$ and $\delta_1^{(i)} - \delta_2^{(i)}$ as $\Delta\delta_i$ in the following.

Assume that $\epsilon \ll d$ for $l_0$, $l_1$ and $l_2$ bounded perturbations, and $\epsilon \ll 1$ for the $l_\infty$ bounded perturbation. Then, $\forall q \geq 1$, we have

$$
\begin{aligned}
l_0 \text{ budget:} \quad & \sum_i |\Delta\delta_i|^q \leq 2\epsilon, \\
l_1 \text{ budget:} \quad & \sum_i |\Delta\delta_i|^q \leq D_1 + (2\epsilon - D_1)^q, \\
l_2 \text{ budget:} \quad & \sum_i |\Delta\delta_i|^q \leq D_2 + (4\epsilon^2 - D_2)^{\frac{q}{2}}, \\
l_\infty \text{ budget:} \quad & \sum_i |\Delta\delta_i|^q \leq d \times (2\epsilon)^q,
\end{aligned}
\tag{39}
$$

where $D_1 = \lfloor 2\epsilon \rfloor$ and $D_2 = \lfloor 4\epsilon^2 \rfloor$. The derived upper bounds are tight because

**(1)** $l_0$ **budget:** The equality achieves when the location of non-zero elements in $\boldsymbol{\delta}_1$ and $\boldsymbol{\delta}_2$ has no overlap, and the magnitude of their non-zero elements reaches $\pm 1$.

**(2)** $l_1$ **budget:** Since $0 \leq |\Delta\delta_i| \leq 1$, the equality achieves when there exists at most one $\Delta\delta_k$ such that $|\Delta\delta_k| < 1$ and $\forall j \neq k, |\Delta\delta_j| = 1$. The maximum number of $\Delta\delta_j$ is $\lfloor 2\epsilon \rfloor$. Then, according to $\|\Delta\boldsymbol{\delta}\|_1 \leq 2\epsilon$, we have $|\Delta\delta_k| = 2\epsilon - 1 \times \lfloor 2\epsilon \rfloor$.

**(3)** $l_2$ **budget:** The derivation is similar to that of the $l_1$ case.

**(4)** $l_\infty$ **budget:** The equality achieves when $\boldsymbol{\delta}_1 = -\boldsymbol{\delta}_2$.

On popular benchmark CIFAR-10, $d = 32 \times 32 \times 3 = 3072$, and the commonly used values of $\epsilon$ in the $l_0$, $l_1$, $l_2$ and $l_\infty$ cases are 360, 24, 0.5 and 8/255, respectively [5, 23, 41, 19]. Substitute these into (39), we can easily get that $\forall q \geq 1$, the upper bound of $\sum_i |\Delta\delta_i|^q$ is significantly larger in the $l_0$ case than the other cases. For instance, $(2\epsilon - D_1)^q$, $(4\epsilon^2 - D_2)^{\frac{q}{2}}$ and $(2\epsilon)^q$ reach their respective maximum values when $q = 1$, since all of them are smaller than 1. Then, the upper bounds of $\sum_i |\Delta\delta_i|^1$ in the $l_0$, $l_1$, $l_2$ and $l_\infty$ cases are 720, 24, 1 and $49152/255 \approx 192.8$, respectively.

Furthermore, the $l_q$ norm of $\Delta\boldsymbol{\delta}$ is defined as follows:

$$\|\Delta\boldsymbol{\delta}\|_q = \left( \sum_i |\Delta\delta_i|^q \right)^{\frac{1}{q}}. \tag{40}$$

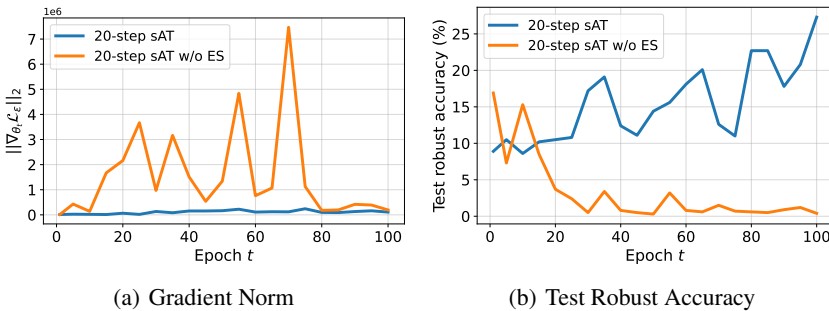

(a) Gradient Norm            (b) Test Robust Accuracy

Figure 4: Relationship between craggy loss landscape and CO. **(a)** Gradient norm $\|\nabla_{\theta_t}\mathcal{L}_\epsilon\|_2$. **(b)** Test robust accuracy against sAA ($\epsilon = 20$). The results are obtained from PreactResNet-18 trained on CIFAR-10 with $\epsilon_{train} = 40$. Since the training of 20-step sAT w/o ES diverges under $\epsilon_{train} = 120$, the results are presented under $\epsilon_{train} = 40$ instead.

Since the upper bound of $\sum_i |\Delta\delta_i|^q$ in the $l_0$ case is larger than 1 for all $q \geq 1$, we can also derive that $\forall q \geq 1$, the upper bound of $\|\Delta\boldsymbol{\delta}\|_q$ is always significantly larger in the $l_0$ case than the other cases.

# F  More Experimental Results

## F.1  Early Stopping in Multi-step Adversarial Training Avoids Catastrophic Overfitting

In Figure 4, we compare the cases with and without ES in terms of gradient norm and robust accuracy on the test set by sAA. We can observe that 20-step sAT without ES still suffers from CO and the corresponding gradient magnitude during training indicates a craggy loss landscape.

## F.2  Distances between Gradients Induced by 1-step and Multi-step Attacks

Table 6: Average $l_2$ distances between gradients induced by 1-step and multi-step attacks, represented by $\|\nabla_{\boldsymbol{\theta}}\mathcal{L}_\epsilon(\boldsymbol{x} + \delta_{one}) - \nabla_{\boldsymbol{\theta}}\mathcal{L}_\epsilon(\boldsymbol{x} + \delta_{multi})\|_2$. The gradients are calculated of the training set of CIFAR-10 [25]. The $l_0, l_1, l_2$ and $l_\infty$ models are obtained by 1-step sAT [23], Fast-EG-$l_1$ [19], 1-step PGD [38] and GradAlign [37], respectively. The 1-step and multi-step $l_0$ attacks are 1-step and 10000-step sPGD [23], respectively. The 1-step and multi-step $l_1$ attacks are 1-step Fast-EG-$l_1$ and 100-step APGD [41], respectively. The 1-step and multi-step attacks for other norms are 1-step PGD [5] and 100-step APGD [3], respectively.

| Model | $l_0$ ($\epsilon = 1$) | $l_1$ ($\epsilon = 24$) | $l_2$ ($\epsilon = 0.5$) | $l_\infty$ ($\epsilon = 8/255$) |
|---|---|---|---|---|
| $l_2$ distance | 15.8 | $9.1 \times 10^{-4}$ | $3.6 \times 10^{-4}$ | $6.7 \times 10^{-4}$ |

Based on the Lipschitz smoothness assumption in Inequality (6), the gradient difference arising from approximated adversarial perturbations is bounded by $L_{\boldsymbol{\theta}\boldsymbol{x}}\|\boldsymbol{\delta}_1 - \boldsymbol{\delta}_2\|$ where $\boldsymbol{\delta}_1$ is the perturbation generated by 1-step attack and $\boldsymbol{\delta}_2$ is the optimal perturbation. Based on the same reason that $l_0$ norm is not a proper norm, $\|\boldsymbol{\delta}_1 - \boldsymbol{\delta}_2\|$ is significantly larger in $l_0$ cases than $l_\infty, l_2$ and $l_1$ cases, which makes 1-step adversarial training more challenging in $l_0$ cases. To corroborate this, we compare the distance between gradients induced by 1-step and multi-step attacks. As presented in Table 6, the average distance between gradients induced by 1-step and multi-step $l_0$ attacks is 5 orders of magnitude greater than those in the $l_1, l_2$ and $l_\infty$ cases, even when a single pixel is perturbed. This finding indicates that the loss landscape of $l_0$ adversarial training is significantly more craggy than other cases in the input space.

## F.3  Comparison with Other Smoothing Approaches

In this section, we undertake a more comprehensive comparison between our proposed Fast-LS-$l_0$ and other smoothing approaches (label smoothing (LS) [54], AdvLC [31], MART [32] and SWA [33]). Note that all baselines are tuned by a thorough hyperparameter search. As demonstrated in Table 7, our method achieves the strongest robustness against sAA. First, naive LS turns out ineffective under

Table 7: Comparison with other smoothing approaches in robust accuracy (%) by sAA. The target sparsity level $\epsilon = 20$. We compare PreAct ResNet-18 [24] models trained on CIFAR-10 [25]. The *italic numbers* indicate catastrophic overfitting (CO) happens.

| Method | LS | AdvLC | MART | TRADES+SWA | **Ours** | **Ours+ SWA** |
|---|---|---|---|---|---|---|
| Robust Acc. | *0.0* | 59.6 | 48.0 | 45.0 | **63.0** | 59.1 |

the $l_0$ setting. The performance of AdvLC, MART, and SWA is not as good as the method we use. Additionally, the combination of our method with SWA harms robustness due to over-smoothing.

## F.4 More Results of Section 5.2

Table 8: Robust accuracy (%) of various models on different attacks that generate $l_0$ bounded perturbations, where the sparsity level $\epsilon = 10$. The models are PreAct ResNet-18 trained on **CIFAR-100** [25] with $\epsilon = 60$. Note that the results of vanilla sAT and sTRADES are obtained from [23], CornerSearch (CS) is evaluated on 1000 samples due to its high computational complexity.

| Model | Time Cost | Clean | Black-Box | | SAIF | White-Box | | sAA |
|---|---|---|---|---|---|---|---|---|
| | | | CS | RS | | $sPGD_{proj}$ | $sPGD_{unproj}$ | |
| *Multi-step* | | | | | | | | |
| sAT | 4h 27m | 67.0 | 44.3 | 41.6 | 60.9 | 56.8 | 58.0 | 41.6 |
| **+S&N** | 4h 58m | 64.3 | 53.0 | 52.9 | 61.2 | 59.2 | 59.6 | 52.8 |
| sTRADES | 5h 10m | 70.9 | 52.8 | 50.3 | 65.2 | 64.0 | 63.7 | 50.2 |
| **+S&N** | 5h 40m | 63.8 | 56.5 | 55.6 | 61.2 | 60.5 | 59.0 | **55.3** |
| *One-step* | | | | | | | | |
| **Fast-LS-$l_0$ (T)** | 1h 05m | 65.3 | 54.5 | 54.3 | 60.4 | 55.6 | 54.4 | 52.2 |
| **Fast-LS-$l_0$ (F)** | 1h 26m | 65.0 | 56.2 | 54.6 | 60.8 | 54.9 | 54.9 | **52.3** |

Table 9: Robust accuracy (%) of various models on different attacks that generate $l_0$ bounded perturbations, where the sparsity level $\epsilon = 12$. The models are PreAct ResNet-18 trained on **GTSRB** [52] with $\epsilon = 72$. All methods are evaluated on 500 samples, and CornerSearch (CS) is not evaluated here due to its high computational complexity.

| Model | Time Cost | Clean | Black-Box | | SAIF | White-Box | | sAA |
|---|---|---|---|---|---|---|---|---|
| | | | CS | RS | | $sPGD_{proj}$ | $sPGD_{unproj}$ | |
| *Multi-step* | | | | | | | | |
| sAT | 1h 3m | 98.4 | - | 43.2 | 92.4 | 96.0 | 96.2 | 43.2 |
| **+S&N** | 1h 2m | 98.4 | - | 77.8 | 97.4 | 96.8 | 95.4 | **77.6** |
| sTRADES | 1h 6m | 97.8 | - | 67.6 | 94.0 | 95.6 | 95.0 | 67.4 |
| **+S&N** | 1h 7m | 95.6 | - | 75.4 | 93.6 | 92.6 | 91.2 | 75.2 |
| *One-step* | | | | | | | | |
| **Fast-LS-$l_0$ (T)** | 7m | 97.8 | - | 75.2 | 89.2 | 74.4 | 74.4 | 63.2 |
| **Fast-LS-$l_0$ (F)** | 9m | 98.6 | - | 80.4 | 94.2 | 75.0 | 79.8 | **67.8** |

The results on CIFAR-100 and GTSRB datasets are presented in Table 8 and 9, respectively. The findings are consistent with those observed in Table 5(a), further validating the effectiveness of the proposed methods across different datasets. In contrast to the settings in [23], we resize the images in GTSRB to $32 \times 32$ instead of $224 \times 224$ and retrain the models from scratch. The model are trained with $\epsilon = 72$ and evaluated for robustness with $\epsilon = 12$. It is important to note that due to the smaller search space resulting from low-resolution images, the attack success rate of the black-box Sparse-RS (RS) under this setting is significantly higher than that reported in [23].

## F.5 Standard Deviation of Robust Accuracy against Sparse-AutoAttack of Table 5(a)

To better validate the effectiveness of our method, we report the standard deviations of robust accuracy against sAA in Table 10. We calculate these standard deviations by running the experiments three times with different random seeds. The configurations are the same as in Table 5(a). It can be

Table 10: Average robust accuracy against sAA [23] obtained from three runs, where the sparsity level $\epsilon = 20$. The variances are shown in brackets. The configurations are the same as in Table 5(a). Note that we do not include the results of vanilla sAT and sTRADES since their results are obtained from [23].

| Model | sAT + S&N | sTRADES + S&N | Fast-LS-$l_0$ (T) | Fast-LS-$l_0$ (F) |
|---|---|---|---|---|
| Acc. | 61.2 ($\pm$ 0.2) | 65.5 ($\pm$ 0.7) | 63.0 ($\pm$ 0.7) | 62.1 ($\pm$ 0.6) |

observed that the fluctuation introduced by different random seeds does not outweigh the performance gain from the evaluated approaches.

## F.6 Robustness under Feature-space Attacks

Table 11: Robust accuracy (%) of cominations of our method under feature-space attacks. The model is the ResNet-18 trained with Fast-LS-$l_0$ on CIFAR-10.

| $\epsilon = 60$ | Clean Acc. | $\sigma$-zero | sPGD$_p$ | sPGD$_u$ | RS | sAA |
|---|---|---|---|---|---|---|
| Fast-LS-$l_0$ | 84.9 | 64.2 | 76.6 | 73.5 | 58.5 | **58.3** |

We train a ResNet with our method in feature space with $\epsilon_{train} = 360$ and evaluate its robustness under different attacks with $\epsilon = 60$. To accommodate the feature-space sparsity, we modify the shape of perturbation mask in sPGD from $(b, 1, h, w)$ to $(b, c, h, w)$. Since RS is unable to generate feature-space perturbations, we set its pixel-space adversarial budget to $\epsilon/c$ to ensure the equivalent feature-space budget as other attacks. As presented in Table 11, our method can also achieve robustness in this setting. In addition, we notice that $\sigma$-zero outperforms sPGD in feature-space attacks. Nevertheless, sAA is still the most comprehensive and reliable robustness evaluation method.

## F.7 Effectiveness in Improving Fast $l_1$, $l_2$ and $l_\infty$ Adversarial Training

Table 12: Robust accuracy (%) of cominations of our method with fast $l_1$, $l_2$, and $l_\infty$ adversarial training methods. The results are obtained through the AutoAttack for corresponding norms.

| | $l_1$ ($\epsilon = 12$) | | | $l_2$ ($\epsilon = 0.5$) | $l_1$ ($\epsilon = 8/255$) |
|---|---|---|---|---|---|
| **Fast-EG-$l_1$** | 45.4 | | **GradAlign** | 60.2 | 39.4 |
| **+ Ours** | **49.2** | | **+ Ours** | **63.1** | **42.4** |

We combine our method with fast $l_1$, $l_2$, and $l_\infty$ adversarial training methods. Specifically, we use Fast-EG-$l_1$ [19] for $l_1$ cases and GradAlign [27] for $l_2$ and $l_\infty$ cases. The results in Table 12 show that our method enhances robustness in all cases. Overall, these findings demonstrate the effectiveness of our method in enhancing robustness against different attacks.

## F.8 Evaluation on Different Networks

Table 13: Robust accuracy (%) of various networks against sAA on CIFASR-10, where the sparsity level $\epsilon = 20$. The networks are adversarially trained with different methods, including 1-step sAT, 1-step sTRADES and the proposed Fast-LS-$l_0$.

| | PRN-18 | ConvNeXt-T | Swin-T |
|---|---|---|---|
| 1-step sAT | 0.0 | 0.8 | 0.1 |
| 1-step sTRADES | 31.0 | 71.0 | 43.2 |
| **Fast-LS-$l_0$** | **63.0** | **78.6** | **58.9** |

Despite the effectiveness of our method on PreActResNet-18 (PRN-18) and ResNet-34, the performance of our Fast-LS-$l_0$ and its ablations on different networks remains unexplored. In this regard, we further evaluate our method on two popular architectures, i.e., ConvNeXt [55] and Swin Transformer [56]. Note that we adopt their tiny versions for CIFAR-10, which have a similar number of parameters as ResNet-18, and we follow the training settings of their CIFAR-10 implementations. The other experimental settings are the same as those described in Section 5.1. As shown in Table

13, vanilla adversarial training results in CO on all networks, and our method produces the best robust accuracy against sAA, demonstrating the effectiveness of our method on different networks. Notably, ConvNeXt shows surprisingly strong robustness against sAA, suggesting that advanced architecture design and dedicated hyperparameter tuning can provide additional performance gains. However, as Transformers has struggled to perform well on small datasets without pretraining [57], Swin Transformer also underperforms CNN-based networks in this scenario.

### F.9 Loss Landscape of one-step sAT with Different $\epsilon$

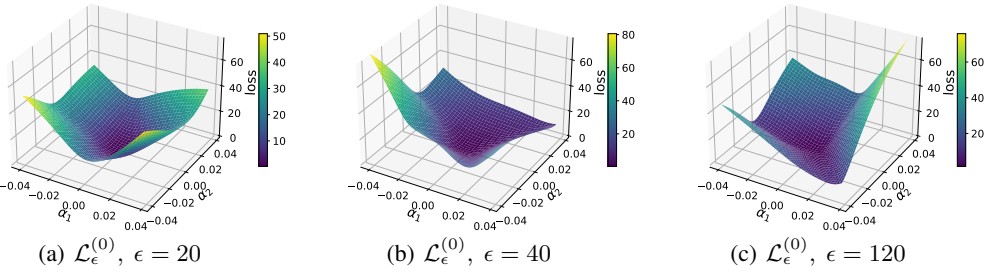

(a) $\mathcal{L}_\epsilon^{(0)}$, $\epsilon = 20$      (b) $\mathcal{L}_\epsilon^{(0)}$, $\epsilon = 40$      (c) $\mathcal{L}_\epsilon^{(0)}$, $\epsilon = 120$

Figure 5: Loss landscape of 1-step sAT [23] with different $\epsilon$ values on the training set of CIFAR-10 [25]. The architecture of the model is PreactResNet-18. **(a)** Landscape of $\mathcal{L}_\epsilon^{(0)}(\boldsymbol{x}, \boldsymbol{\theta} + \alpha_1 \boldsymbol{v}_1 + \alpha_2 \boldsymbol{v}_2)$ with $\epsilon = 20$, where $\boldsymbol{v}_1$ and $\boldsymbol{v}_2$ are the eigenvectors corresponding to the top 2 eigenvalues of the Hessian matrices, respectively. **(b)** Landscape of $\mathcal{L}_\epsilon^{(0)}$ with $\epsilon = 40$. **(c)** Landscape of $\mathcal{L}_\epsilon^{(0)}$ with $\epsilon = 120$.

As supplementary of Figure 3, we visualize the loss landscapes of 1-step sAT [23] with different $\epsilon$, including 20, 40 and 120, in Figure 5. It can be observed that the $l_0$ adversarial loss exhibits a drastic increase in response to relatively minor alterations in the $\boldsymbol{\theta}$-space. Moreover, the degree of non-smoothness increases in proportion to $\epsilon$, which is consistent with the observation in Figure 3 (a).

### F.10 Smoother Loss Landscape Induced by Soft Label and Trade-off Loss Function

The effectiveness of soft label and trade-off loss function in improving the performance of $l_0$ adversarial training is demonstrated in Section 5.1 and 5.2. Additionally, we visualize the curves of top-10 eigenvalues of Hessian matrices of the different methods discussed in Section 5.1 and their respective loss landscapes in Figure 6. Note that since N-FGSM results in a larger upper bound of $\|\boldsymbol{\delta}_1 - \boldsymbol{\delta}_2\|$, it is not considered here to make a fair comparison. Figure 6 (a) shows that sTRADES induces considerably smaller eigenvalues of Hessian matrices compared to sAT, while the difference between sTRADES (T) and sTRADES (F) is negligible. SAT, on the other hand, has only a marginal effect on the eigenvalues. However, as illustrated in Figure 6 (b)-(f), SAT plays a crucial role in smoothing the loss landscape, which relates to the change rate of loss, i.e., the first-order smoothness. These observations align with the theoretical derivation presented in Section 4.3, indicating that soft label improves the first-order smoothness, while trade-off loss function contributes to the second-order smoothness.

### F.11 Ablation Studies

In this section, we conduct more ablation studies on the results in Section 5.1. Specifically, we focus on the best configuration in Table 4: Fast-LS-$l_0$ (T) (i.e., 1-step sTRADES (T) + SAT & N-FGSM). Unless specified, we adopt the same training settings as in Table 4.

Table 14 presents a performance comparison of the model when SAT is enable in different training phases. We can see that the performance achieves the best when enabling SAT at the 50-th epoch. This observation demonstrates that the best performance in 1-step sTRADES is achieved when SAT is enabled at the intermediate epoch where the learning rate is relatively low.

In Table 15, we compare the performance when using different momentum factor in SAT. We can see that the default setting in [40], i.e., 0.9, provides the best performance.

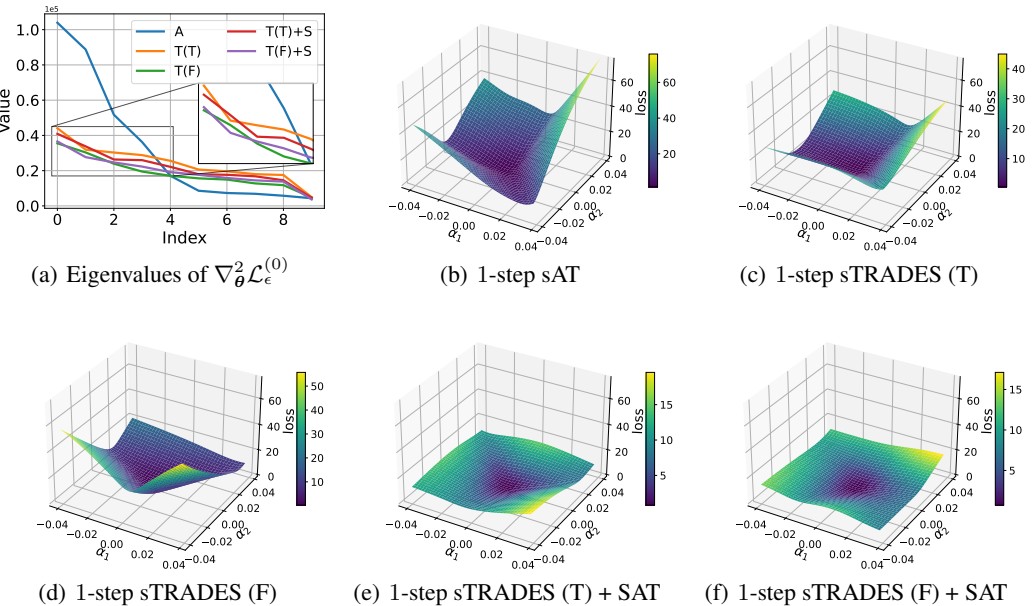

(a) Eigenvalues of $\nabla_{\boldsymbol{\theta}}^2 \mathcal{L}_\epsilon^{(0)}$      (b) 1-step sAT      (c) 1-step sTRADES (T)

(d) 1-step sTRADES (F)      (e) 1-step sTRADES (T) + SAT      (f) 1-step sTRADES (F) + SAT

Figure 6: Smoothness visualization of different methods with $\epsilon = 120$ on the training set of CIFAR-10 [25]. The architecture of the model is PreactResNet-18. **(a)** Top-10 eigenvalues of $\nabla_{\boldsymbol{\theta}}^2 \mathcal{L}_\epsilon^{(0)}(\boldsymbol{x}, \boldsymbol{\theta})$ of different methods. A and T denote 1-step sAT and 1-step sTRADES, respectively. T and F in the brackets are two respective versions of sTRADES indicated in Sec. 5.1. **(b)** Loss landscape of 1-step sAT. **(c)** Loss landscape of 1-step sTRADES (T). **(d)** Loss landscape of 1-step sTRADES (F). **(e)** Loss landscape of 1-step sTRADES (T) + SAT. **(f)** Loss landscape of 1-step sTRADES (F) + SAT.

In Table 16, we compare the performance when using different balance factor $\beta$ in TRADES. It can be observed that $\beta = 3$ and 6 induce similar results, indicating the default setting in [30], i.e., 6, is the optimal.

In Table 17, we compare the performance when using different $\epsilon_{train}$. It can be observed that when $\epsilon_{train} = 120$, our method achieves the best performance, which is consistent with the observations in [23]. Notably, with the assistance of our method, CO does not appear in all settings.

Table 14: Ablation study on the epoch of enabling SAT. The evaluated attack is sAA, where the sparsity level $\epsilon = 20$.

| SAT epoch | 30 | 50 | 70 |
|---|---|---|---|
| Robust Accuracy | 60.2 | **63.0** | 62.8 |

Table 15: Ablation study on the momentum factor of SAT. The evaluated attack is sAA, where the sparsity level $\epsilon = 20$.

| SAT momentum | 0.5 | 0.7 | 0.9 |
|---|---|---|---|
| Robust Accuracy | 55.4 | 60.4 | **63.0** |

Table 16: Ablation study on the balance factor $\beta$ in TRADES loss function. The evaluated attack is sAA, where the sparsity level $\epsilon = 20$.

| TRADES $\beta$ | 1 | 3 | 6 |
|---|---|---|---|
| Robust Accuracy | 58.7 | **63.0** | **63.0** |

Table 17: Ablation study on $\epsilon_{train}$. The evaluated attack is sAA, where the sparsity level $\epsilon = 20$.

| $\epsilon_{train}$ | 20 | 40 | 120 |
|---|---|---|---|
| Robust Accuracy | 61.4 | 62.1 | **63.0** |

# G    Implementation Details

Generally, the epoch of enabling SAT is $1/2$ of the total epochs. For N-FGSM, the random noise for augmentation is the random sparse perturbation with sparsity level ranging from 0 to $2\epsilon$, where $\epsilon$ is the sparsity level of adversarial perturbations. The interpolation factor $\alpha$ in trade-off loss function is set to 0.75. The balance factor $\beta$ in TRADES loss function is set to 6. The optimizer is SGD with a momentum factor of 0.9 and a weight decay factor of $5 \times 10^{-4}$. The learning rate is initialized to

0.05 and is divided by a factor of $10$ at the $1/4$ and $3/4$ of the total epochs. The specific settings for different datasets are listed as follows:

- **CIFAR-10, CIFAR-100 [25] and GTSRB [52]:** The adopted network is PreAct ResNet-18 [58] with softplus activation [59]. The training batch size is $128$. We train the model for $100$ epochs.
- **ImageNet-100 [51]:** The adopted network is ResNet-34 [24]. The training batch size is $48$. We train the model for $50$ epochs.

Unless specified, the hyperparameters of attacks and other configurations are the same as in [23].

