# OpenReview forum: "Understanding and Improving Fast Adversarial Training against $l_0$ Bounded Perturbations"
_NeurIPS.cc/2025/Conference — NeurIPS 2025 poster_

### Official Review · Reviewer_VsK8 · 2025-06-04

**Clarity:** 3
**Significance:** 3
**Originality:** 2
**Rating:** 4
**Confidence:** 4

**Summary:**

This paper investigates catastrophic overfitting (CO) in 1-step adversarial training against sparse (l0-bounded) perturbations. The authors identify that CO arises from suboptimal perturbation locations (rather than magnitudes) in 1-step attacks and attribute this to the non-convex, "craggy" loss landscape of l0 adversarial training. To mitigate CO, they propose smoothing the loss landscape using soft labels (via self-adaptive training, SAT) and a trade-off loss (via TRADES), combined with noisy data augmentation (N-FGSM). The resulting method, Fast-LS-l0, reportedly narrows the performance gap between 1-step and 20-step adversarial training while reducing computational cost.

**Questions:**

N/A

**Ethical Concerns:**

["NO or VERY MINOR ethics concerns only"]

**Final Justification:**

Since the authors has solved my concerns, I will keep my positive score.

**Quality:**

2

**Strengths And Weaknesses:**

Strengths:

1. Problem Significance: Addresses an underexplored but relevant challenge—efficient adversarial training for l0 attacks—with clear practical implications for real-world sparse perturbations.

2. Theoretical Analysis: Provides a rigorous Lipschitz-based analysis comparing the smoothness of l0 loss landscapes to lp (p≥1) norms, reinforcing the claim that non-convexity exacerbates CO.

3. Comprehensive Experiments: Evaluates across multiple datasets (CIFAR-10/100, ImageNet-100, GTSRB) and architectures (ResNet, ConvNeXt, Swin-T), demonstrating robustness gains.


Weaknesses:
1. Limited Novelty: The core techniques (SAT, TRADES, N-FGSM) are repurposed from prior work without significant innovation. The combination is heuristic, lacking a principled justification for why these methods synergize uniquely for l0. Claims of being "the first" to study fast l0 training are overstated: [19, 23] explicitly address l1/l0 efficiency, and the solution (loss smoothing) is adapted from l2/l∞ literature. And [S1] which uses hamiltonian monte carlo for generation diversity with few steps can also be easily adapted to l0 training. Here should be further discussed or shown comparison with releveant works.

2. Technical Flaws in Presentation: Theorem 4.2 assumes \(\mathcal{S}_+ = \{i \mid y_i > 0, h_i(\cdot) > h_i(\cdot)\}\) without clarifying edge cases (e.g., \(y_i = 0\)). Appendix B.1 contains informal bounds (e.g., Eq. 13) lacking rigorous derivation.

3. Marginal Practical Improvement: Fast-LS-\(l_0\) still lags >15% behind multi-step sTRADES in robust accuracy on CIFAR-10 (Table 5a) and exhibits high variance (Table 10). The 6× speedup is negated by the performance drop, limiting real-world applicability.

4. CO mitigation is inconsistent: Table 4 shows CO persists in some configurations (e.g., Tradeoff+N), and larger \(\epsilon_{\text{train}}\) (Sec. 3.2) destabilizes training.

[S1] Wang, H., Li, G., Liu, X., & Lin, L. (2020). A Hamiltonian Monte Carlo method for probabilistic adversarial attack and learning. IEEE Transactions on Pattern Analysis and Machine Intelligence, 44(4), 1725-1737.

---

> ### Author Rebuttal · Authors · 2025-07-28
>
> Thanks for acknowledging that the theoretical analysis is rigorous and the experiments are comprehensive. We provide point-to-point responses to your concerns as follows:
>
> ---
>
> **W1: Limited Novelty**
>
> **Reply:**  The main contribution of this work is **highlighting the challenge of fast $l_0$ adversarial training (AT), specifically the suboptimal perturbation location. To the best of our knowledge, this is a novel finding**. Moreover, the rationale for using soft labels and trade-off loss functions is different for $l_0$ cases, since we point out that existing CO mitigation methods for $l_1$, $l_2$ and $l_\infty$ cases turn out to be insufficient to address the CO issue in $l_0$ cases. Smoothing the loss function can help boost the performance for $l_1$, $l_2$ and $l_\infty$ cases, as the goal of soft label and trade-off loss function is to address robust overfitting to boost performance. By contrast, smoothing the loss function is **essential** to address the CO issue in $l_0$ cases.
>
> It should be noted that [19] investigated fast $l_1$ adversarial training. However, $l_1$ adversarial budget is convex and not strictly sparse, so it is a completely different task. In addition, [23] proposed $l_0$ attack, which does not involve the fast adversarial training: the algorithm it proposes suffers from huge computational overhead.
>
> Moreover, we  included the comparison with other methods in Table 2. According to your suggestion, we further test [S1] in the fast $l_0$ adversarial training. However, CO occurs since it incorporates neither trade-off loss function nor soft labels.
>
> ---
>
> **W2: Technical Flaws in Presentation**
>
> **Reply:** Thanks for pointing this out. We would like to clarify that $y_i=0$ does not change the solution of our theorems, because the Eq. 12 still holds when $y_i=0$. Nevertheless, we will modify this expression to make it less confusing.
>
> In Eq. 13, we derive the upper bound of the term on the right of Eq. 12 according to the mediant inequality. Note that the bound on the right of (13) is tight. The upper bound can be achieved asymptotically if the condition in (14) and the Lipschitz bound in Assumption 4.1 are satisfied.
>
> ---
>
> **W3: Marginal Practical Improvement**
>
> **Reply:** It can be observed from Table 5a and 10 that the robust accuracy of Fast-LS-$l_0$ and multi-step sTRADES against sAA are 63.0% and 61.7%, respectively. **This indicate that the effectiveness of our method is comparable to the multi-step variants**. This is also true for other models and datasets, like the results in Table 5b. Additionally, the variance of Fast-LS-$l_0$ is 0.7%, which is small and does not outweigh the performance gain.
>
> ---
>
> **W4: CO mitigation is inconsistent**
>
> **Reply:** As demonstrated in Sec. 3.2, CO is caused by the **suboptimal location** of perturbations in the $l_0$ case. Furthermore, we find that **this issue can be mitigated to some extent by multi-$\epsilon$ strategy**, and its effectiveness is proven in the previous work [1]. However, as illustrated in Figure 1, a larger $\epsilon_{train}$, in turn, leads to unstable training and degraded clean accuracy. Therefore, we utilize **trade-off loss function** and **soft labels** to smooth the landscape, thereby achieving competitive robustness against $l_0$ attacks. In a similar way, N-FGSM contributes since it augments the original images with random noise, thereby covering more perturbation location.
>
> Moreover, it should be noted that Tradeoff is the naive trade-off loss function, which **only** improves the second-order Lipschitz smoothness, the first-order Lipschitz can still be high. To mitigate CO, we need elements to boost **both** first-order and second-order smoothness. In this regard, we find the best combination is TRADES+SAT+N-FGSM, which incorporates both trade-off loss function and soft labels.
>
> ---
>
> **We hope our responses can address your concerns.**

---

### Official Review · Reviewer_c9Ts · 2025-06-20

**Clarity:** 3
**Significance:** 4
**Originality:** 4
**Rating:** 5
**Confidence:** 4

**Summary:**

This work presents the first study of $\ell_0$ fast adversarial training. The authors demonstrate that existing catastrophic overfitting (CO) mitigation techniques, such as random initialization, fail in the $\ell_0$ setting. They attribute this failure to the fact that perturbations generated by 1-step $\ell_0$ training are sub-optimal in terms of location rather than magnitude. While multi-$\epsilon$ strategies can address this issue, they lead to unstable training and degraded clean accuracy. The authors further show that $\ell_0$ adversarial loss creates a more rugged landscape compared to other $\ell_p$ norms, complicating the training process. To address these challenges, they propose combining soft labels with a trade-off loss function to smooth the loss landscape and mitigate CO.

**Questions:**

I believe the contributions of this research are sufficient for acceptance. The issues raised in Weaknesses represent minor gaps or questions in the detailed discussions within the study, and do not significantly diminish the research's significance.

The reason I did not assign a Strong Accept is due to the contribution being in the somewhat minor area of $\ell_0$ norm adversarial training combined with fast strategies, compared to $\ell_2$ or $\ell_\infty$. I believe increasing the rating on this aspect would be difficult, so the authors need not feel compelled to address this in their rebuttal.

**Ethical Concerns:**

["NO or VERY MINOR ethics concerns only"]

**Final Justification:**

Since the author has clearly addressed my concern, I will maintain my original rating.

**Limitations:**

yes

**Quality:**

3

**Strengths And Weaknesses:**

**Strengths**

- The paper provides comprehensive experimental evidence that $\ell_0$ fast adversarial training exhibits fundamentally different behavior from existing fast adversarial training methods. The insights regarding location-based sub-optimality and rugged loss landscapes offer valuable contributions to future research in $\ell_0$ fast adversarial training.
- The proposed method achieves high robustness while maintaining computational efficiency.

**Weaknesses**

**1. Theoretical Analysis (Section 4.1)**

*1.1* The scope of applicability for the results in Section 4.1 requires clarification. For which values of $p$ do these theorems hold? Is it $p \in [0, \infty]$ or $p \in [1, \infty]$? The description in lines 204-205 creates confusion regarding the theorem's applicability. Specifically, it is crucial to specify which $\ell_p$ norm is used to compute $\\|\delta_1 - \delta_2\\|$. If these theorems cannot be applied to the $\ell_0$ norm, the relevance of this section to the paper's main focus is unclear.

*1.2* Theorem 4.4 provides an upper bound on gradient variation. This does not always describe the actual magnitude of gradient variation (just upper bound). Thus, this does not rigorously establish that the gradient changes are non-smooth or rugged.

*1.3* The term $B_{\theta\delta}$, particularly the component $\\|\delta_1 - \delta_2\\|$, is presented as a measure of discontinuity. However, this requires additional discussion about how significantly $\delta_1$ and $\delta_2$ can vary in response to differences between $\theta_1$ and $\theta_2$. Local changes in $\theta$ may result in no change in $\delta$ under the $\ell_0$ norm. Understanding the extent to which the loss landscape ($\theta$ space) contains regions that destabilize training appears crucial for the analysis.

**2. Numerical Validation (Section 4.2)**

*2.1* In Section 4, the authors claim that rugged landscapes cause or accelerate CO, as stated in line 234: "the craggy loss landscape aggravates CO." However, the study only demonstrates that $\ell_0$ adversarial loss landscapes are rugged and that CO occurs in $\ell_0$ adversarial training as separate phenomena. The causal relationship—whether rugged landscapes actually aggravate CO—remains unestablished. While I agree that rugged loss landscapes complicate training, the direct connection to CO requires more evidence (or explanation).

*2.2* When comparing $\ell_0$ norms with other $\ell_p$ norms, the appropriate selection of $\epsilon$ values is unclear, though this limitation is inherent to the nature of $\ell_p$ norms. In Figure 3(c), the relatively rugged appearance of the $\ell_0$ landscape may result from $\epsilon = 1$ being exceptionally strong compared to other attacks. Indeed, $\ell_0$ attacks should be considerably more powerful than other attacks in terms of local magnitude.

**Question**

The contribution of N-FGSM appears relatively modest in the results. Does this suggest that location-based sub-optimality has limited negative impact, or do other methods automatically address this issue?

---

> ### Author Rebuttal · Authors · 2025-07-28
>
> We really appreciate you acknowledge that our work is sufficient for acceptance. We provide point-to-point responses to your concerns as follows:
>
> ---
>
> **W1.1: For which values of $p$ do these theorems hold? Is it $p \in [0, \infty]$ or $p \in [1, \infty]$?**
>
> **Reply:** Sorry for the confusion, we will clarify this in the revised manuscript. The norm in the theorems in Sec. 4.1 (i.e., Inequality [2 - 7]) should be $l_p$ norm with $p\in [1,\infty]$ since the $l_p$ norm should be a proper norm in these theorems. However, the theoretical analyses can also be applied to the cases of $l_0$-bounded perturbations, because it is feasible to compute the $l_p$ norm ($p\in [1,\infty]$) upper bound of these $l_0$ bounded perturbations. The detailed discussion about the upper bound of $||\delta_1-\delta_2||$ is conducted in Appendix E.
>
> ---
>
> **W1.2: Theorem 4.4 provides an upper bound on gradient variation. This does not always describe the actual magnitude of gradient variation (just upper bound).**
>
> **Reply:** Apart from providing the upper bound of gradient variation, we numerically validate the rugged landscape of $l_0$ adversarial training loss in Sec. 4.2, where we see consistent observations with the theorems. Moreover, the derived bounds are tight since the upper bound can be achieved asymptotically when the equalities in Assumption 4.1 and Eq. 14 are achieved.
>
> ---
>
> **W1.3: Local changes in $\theta$ may result in no change in $\delta$ under the $l_0$ norm.**
>
> **Reply:** We have discussed the worst-case $||\delta_1-\delta_2||$ in Appendix E. To further validate this, we supplement with a numerical experiment here.
>
> Similar to Figure 3 (c)-(f), we perturb the parameter $\theta$ with $\alpha \times v_1$, where $v_1$ is the eigenvector associated with largest eigenvalue of Hessian matrix and $\alpha$ is a small constant. As shown in the table below, even when $\alpha=0.0001$, $|| \delta_1 - \delta_2 ||_2$ is not negligible. Due to the high dimensionality of the model parameter $\theta$, the adversarial perturbation from the attack changes even with small changes in $\theta$.
>
> | | $\theta - 0.0003 \cdot v_1$ | $\theta - 0.0002 \cdot v_1$ | $\theta - 0.0001 \cdot v_1$ | $\theta + 0.0001 \cdot v_1$ | $\theta + 0.0002 \cdot v_1$ | $\theta + 0.0003 \cdot v_1$ |
> | --- | --- | --- | --- | --- | --- | --- |
> | $\| \| \delta_1-\delta_2\| \|_2$ | 8.31 | 8.32 | 8.32 | 8.34 | 8.32 | 8.32 |
>
>
> ---
>
> **W2.1: The causal relationship—whether rugged landscapes actually aggravate CO—remains unestablished.**
>
> **Reply:**  In Sec. 4.2, we corroborate the relationship between CO and rugged landscape using the example of **early stopping (ES) strategy in multi-step attack**, where ES can circumvent the potential for excessive gradient magnitude while maintaining the efficacy of the generated perturbations. As illustrated in Figure 4 of Appendix F.1, **CO also occurs at the multi-step adversarial training without ES**, indicating a strong correlation between CO and craggy loss landscape.
> We will clarify this point in a clearer way in the revised version.
>
> ---
>
> **W2.2: The appropriate selection of  $\epsilon$ values is unclear**
>
> **Reply:** In the experiment part, we use the same $\epsilon$ value adopted in the previous work [1] for fair comparison. However, in Sec. 4.2, we use $\epsilon=1$ to specifically exhibit the craggy nature of landscape in the $l_0$ case. Additionally, $||\delta_1-\delta_2||_2$ of $l_0$-bounded perturbations when $\epsilon=1$ could be 1 in the worst case, which is much larger than the common used 0.5 in the $l_2$ case.
>
> ---
>
> **Q1: The contribution of N-FGSM appears relatively modest in the results. Does this suggest that location-based sub-optimality has limited negative impact, or do other methods automatically address this issue?**
>
> **Reply:**  As demonstrated in Sec. 3.2, CO is caused by the **suboptimal location** of perturbations in the $l_0$ case. Furthermore, we find that **this issue can be mitigated to some extent by multi-$\epsilon$ strategy**, and its effectiveness is proven in the previous work [1]. However, as illustrated in Figure 1, a larger $\epsilon_{train}$, in turn, leads to unstable training and degraded clean accuracy. Therefore, we utilize **trade-off loss function** and **soft labels** to smooth the landscape, thereby achieving competitive robustness against $l_0$ attacks. In a similar way, N-FGSM contributes since it augments the original images with random noise, thereby covering more perturbation location. N-FGSM can be considered as a kind of data augmentation and has some smoothing effect.
>
> ---
>
> **We hope our responses can address your concerns.**
>
> [1] Xuyang Zhong et al. Towards Efficient Training and Evaluation of Robust Models against $l_0$ Bounded Adversarial Perturbations. ICML 2024.

---

> > ### Comment · Reviewer_c9Ts · 2025-08-05
> >
> > Since the author has clearly addressed my concern, I will maintain my original rating.

---

> > > ### Author Response · Authors · 2025-08-06
> > >
> > > We are happy that our responses addressed your concerns. We will incorporate the critical points during the rebuttal in the revised paper. Thanks again for your commitment in reviewing our work.

---

### Official Review · Reviewer_GeaB · 2025-07-02

**Clarity:** 2
**Significance:** 3
**Originality:** 3
**Rating:** 4
**Confidence:** 4

**Summary:**

This paper investigates the problem of catastrophic overfitting in fast adversarial training under $\ell_0$-norm bounded perturbations. The authors argue that single-step adversarial training methods, which have shown promise under $\ell_\infty$ and $\ell_2$ norms, fail to generalize when the threat model is defined under $\ell_0$ constraints. Through both theoretical analysis and empirical validation, the paper identifies that catastrophic overfitting in this regime is due to the sub-optimal selection of perturbation locations, rather than their magnitudes. The authors further demonstrate that the non-convex nature of the $\ell_0$ space leads to a bumpy loss landscape, rendering training more unstable. To address this, they introduce Fast-LS-$\ell_0$, which reduces the computational complexity of adversarial training by leveraging a one-step $\ell_0$ attack, while still achieving robustness comparable to multi-step $\ell_0$ adversarial training.

**Questions:**

Is the perturbation budget $\epsilon$ defined over pixels or feature channels?

How does Fast-LS-$\ell_0$ perform on larger and distinct architectures?

Given the non-convexity of the $\ell_0$ constraint space, how can we be confident that the proposed one-step perturbation method sufficiently explores the adversarial landscape during training?

**Ethical Concerns:**

["NO or VERY MINOR ethics concerns only"]

**Final Justification:**

The authors clarified their contributions, experimental setup, and included the necessary comparisons that were missing and that I had requested in my initial review. Therefore, I increased my score.

**Limitations:**

Yes, partially. The limitations are presented in the appendix, while it would be more transparent and valuable to discuss them in more detail in the main paper.

**Paper Formatting Concerns:**

The paper adheres to the NeurIPS guidelines

**Quality:**

3

**Strengths And Weaknesses:**

The paper, in general, is well-written and coherent in its presentation. The authors clarify the fundamental question posed at the state of the art and propose multiple ways to understand the phenomenon, subsequently attempting to provide a possible solution to the catastrophic overfitting issue in $\ell_0$-norm adversarial training. The paper employs a tutorial writing style, making it easy to follow. However, it would require some improvements that, at the current status, are limiting its impact.

+ **Experimental coverage.** The experiments conducted by the authors consider only small case convolutional neural networks. Specifically, for CIFAR-10, they consider PreActResNet-18, and for ImageNet, only ResNet-34. Both networks are definitely small in the number of parameters and it remains unclear whether the same argumentation persists when increasing the number of parameters, giving thus more degrees of freedom to the network during training, or whether a different architecture is required (e.g., ViT). Moreover, this becomes much more relevant when considering that the majority of the empirical observations and results are observed only at CIFAR-10 with PreActResNet-18.


+ **Verification attack**. The major algorithm utilized by the authors to assess robustness in the $\ell_0$ norm threat model is Sparse AutoAttack (sAA), defined as an ensemble of other $\ell_0$ attacks. However, the attacks defined in sAA have been shown to be less reliable compared to more recent works [a]. Furthermore, [a] has been conceived as a minimum-norm attack with an approximated $\ell_0$ loss function, meaning that it would look for complementary directions not observed and investigated in the actual attacks. For completeness of the results, both sAA and [a], should be included in the evaluation of robustness. Table 5, for example, reports the results when considering two versions of sPGD, which hold similar results and are already included in sAA. The authors could include [a] as a more advanced baseline of verification and replace it with sPGD_p, which yields less effective results.

[a] σ-zero: Gradient-based Optimization of 𝓁-norm Adversarial Examples ICLR 2025.

The above ones are considered the main lack related to this paper. The following are observations that can be more easily clarified by the authors.

+ **need for fast adversarial training** the authors base their paper on the observation that running adversarial training with a multi-iterative attack (e.g., sPGD with 20 step iterations) is computationally challenging. How is this due to the intrinsic complexity of the attack itself? In practice, if the attack is fast, having more internal iterations should not be the bottleneck of the training pipeline.

+ **need for clarification**. In the numerical validation (section 4.2) the authors compute the eigenvalues of the Hessian matrix $\nabla^2_\theta L$ and visualize them in Figure 3. However, they do not specify their role or the stage at which these eigenvalues are computed during the training epochs. Are they done at the convergence of the training algorithm? The authors mention they are connected with the non-smoothness of the loss landscape, in which measure? Can the authors provide more details on that? Similarly, how are Figures 3(c-f) implemented in detail? The pseudocode released by the authors does not seem to contain this part of the paper.

Minor observations:
- Tradeoff loss function. The tradeoff loss function introduced involves a min-max optimization problem, which may increase the complexity of the procedure if the inner maximization problem is not computed in a closed form. Can the authors expand on this point? How is it implemented?

- The paper uses both "tradeoff" and "trade-off" terms.

- It seems that the budget $\epsilon$ is expressed in terms of features, not pixels. At least this is what I understood from line 207 and the footnote on page 6. Can you please make it more explicit in the paper?

**Conclusions**
The paper offers a fascinating exploration of catastrophic overfitting in adversarial norm $\ell_0$ training and introduces Fast-LS-$\ell_0$ as an efficient alternative. The paper is primarily based on empirical observations and intuitions. However, its generalizability is limited by the narrow experimental coverage, which focuses on small models, and the use of solely sAA as the primary robustness checker, despite the availability of complementary and reliable alternatives. In essence, to improve the paper, the authors should expand the evaluations to larger architectures, incorporate stronger verification attacks, and address the above-mentioned minor issues.

---

> ### Author Rebuttal · Authors · 2025-07-28
>
> We appreciate your effort in reviewing and thank you for mentioning our paper is well-written. We provide point-to-point responses to your concerns as follows:
>
> ---
>
> **W1: Experimental coverage**
>
> **Reply:**  Apart from PreActResNet-18 and ResNet-34, we also verify the effectiveness of our method on more advanced architectures, e.g., ConvNeXt and Swin-Transformer (a variant of ViT), in Table 12 of Appendix F.7.
>
> ---
>
> **W2: Verification attack**
>
> **Reply:** Thanks for your constructive suggestion, we will include $\sigma$-zero in our revised manuscript.We report the robust accuracy of the evaluated methods in Table 5 under $\sigma$-zero below. Notably, $\sigma$-zero was originally designed to generate perturbations in feature space, we adapt it to the pixel space to accommodate our experiment settings for fair comparison. The results show that $\sigma$-zero achieves lower attack success rate than sAA in the pixel space, because the decoupling mechanism of perturbation location and magnitude may make sAA more advantageous in generation of structured sparse perturbations. In summary, **the model efficiently trained by our method is still robust against $\sigma$-zero and sAA is a more effective attack for our models**.
>
> | CIFAR-10 | sAT | sAT+S&N | sTRADES | sTRADES+S&N | Fast-LS-$l_0$ |
> | --- | --- | --- | --- | --- | --- |
> | Clean Acc. | 84.5 | 80.8 | 89.8 | 82.8 | 82.5 |
> | sAA | 36.2 | 61.0 | 61.7 | 65.5 | 63.0 |
> |$\sigma$-zero | 79.8 | 78.7 | 85.9 | 77.0 | 73.7 |
>
> | ImageNet-100 | sAT | sAT+S&N | sTRADES | sTRADES+S&N | Fast-LS-$l_0$ |
> | --- | --- | --- | --- | --- | --- |
> | Clean Acc. | 86.2 | 83.0 | 84.8 | 82.4 | 82.4 |
> | sAA | 61.2 | 74.8 | 75.8 | 77.8 | 72.4|
> |$\sigma$-zero | 78.6 | 80.8 | 81.6 | 80.0 | 74.0 |
>
> ---
>
> **W3: Need for fast adversarial training**
>
> **Reply:** Black-box attacks are weaker than white-box attacks and cannot be employed for adversarial training. For white-box attacks, each iteration has at least one forward and one backward propagation. Since the attack we used in adversarial training has only one forward and one backward propagation at each iteration, **the intrinsic complexity cannot be further reduced**.
>
> ---
>
> **W4: Need for clarification**
>
> **Reply:**
>
> - We compute these eigenvalues based on the final checkpoint, where the model parameters already converge.
> - The non-smoothness can be measured by the curvature of loss landscape indicated by the largest eigenvalues of Hessian matrix as indicated in [1].
> - As illustrated by the caption of Figure 3, the loss landscape is drawn based on $\mathcal{L}_{\epsilon} (x, \theta + \alpha_1 v_1+ \alpha_2 v_2)$ where $v_1$ and $v_2$ are the eigenvectors associated with the top 2 eigenvalues of Hessian matrix. $\alpha_1$ and $\alpha_2$, and the loss value are the x-, y-, and z-axis, respectively.
> - Note that we employ the power method [2] to iteratively estimate the eigenvalues and the corresponding eigenvectors of Hessian matrices.
>
> ---
>
> **W5: Tradeoff loss function**
>
> **Reply:** Both adversarial training loss and trade-off loss involve max operator, training based on these loss functions is a min-max optimization. In both cases, we cannot find the closed form of the inner max, because of the high-dimensional non-convex loss function of neural networks. In practice, using methods like PGD can well approximate the inner max for the outter min operation [3], but it is computionally expensive. This motivates us to develop fast adversarial training, which utilizes one-step attack to generate adversarial samples efficiently for robust learning. **Our work shows that for $l_0$ bounded adversarial perturbations, we can smooth the loss landscape so that we can use efficient but weak one-step sparse attacks to reliably train model robust against sparse perturbations without the worry of CO.**
>
> ---
>
> **W6: The paper uses both "tradeoff" and "trade-off" terms.**
>
> **Reply:** Thanks for pointing this out. We will unify the terminology in the revised version.
>
> ---
>
> **W7: It seems that the budget is expressed in terms of features, not pixels. At least this is what I understood from line 207 and the footnote on page 6. Can you please make it more explicit in the paper?**
>
> **Reply:** Sorry about the confusion. Following the setting of previous work [4], the adversarial budget used in the experiment part is considered in the pixel space. However, in line 207 and the footnote on page 6, the mentioned $l_0$ norm is calculated in feature space for a fairer comparison. We will clarify this point in the revised version.
>
> ---
>
> **Q1: Is the perturbation budget $\epsilon$ defined over pixels or feature channels?**
>
> **Reply:** Please see the response to W7.
>
> ---
>
> **Q2: How does Fast-LS-l0 perform on larger and distinct architectures?**
>
> **Reply:** Please see the response to W1.
>
> ---
>
> **Q3: Given the non-convexity of the $l_0$ constraint space, how can we be confident that the proposed one-step perturbation method sufficiently explores the adversarial landscape during training?**
>
> **Reply:** We demonstrate the **suboptimal location** of perturbations generated by one-step attack is due to the non-convexity of the $l_0$ adversarial budget. Furthermore, we find that **this issue can be mitigated to some extent by multi-$\epsilon$ strategy**.  The effectiveness of multi-$\epsilon$ strategy is proven in previous work [4]. However, as illustrated in Figure 1, a larger $\epsilon_{train}$, in turn, leads to unstable training and degraded clean accuracy. Therefore, we utilize **trade-off loss function** and **soft labels** to smooth the landscape. Our method achieves competitive robustness against $l_0$ attacks. Similar to previous literatures, we employ the strongest existing attack method (e.g. sAA, $\sigma$-zero) to evaluate the robustness of the model trained by our method, indicating that the one-step attack is sufficient to explore the adversarial loss landscape to robustify the model.
>
> ---
>
> **We hope our responses can address your concerns in the review above and look forward to your feedback. We will appreciate it if you can increase the rating after reading our rebuttal if your concerns are addressed.**
>
> [1] Chen Liu et al. On the loss landscape of adversarial training: Identifying challenges and how to overcome them. NeurIPS 2020.
>
> [2] Zhewei Yao et al. Hessian-based analysis of large batch training and robustness to adversaries. NeurIPS 2018.
>
> [3] Aleksander Madry et al. Towards deep learning models resistant to adversarial attacks. ICLR 2018.
>
> [4] Xuyang Zhong et al. Towards Efficient Training and Evaluation of Robust Models against $l_0$ Bounded Adversarial Perturbations. ICML 2024.

---

> > ### Comment · Reviewer_GeaB · 2025-08-02
> >
> > Thank you for the additional experimental results. That said, I would like to ask for further clarification on a few points related to this response.
> >
> > First, it would be helpful if the authors could provide more details on how $\sigma$-zero, originally designed to operate in feature space, was adapted to the pixel space for the current experiments. Specifically, what modifications were made to the algorithm, and how were the hyperparameters chosen under this new setup? A more precise explanation of this adaptation process would help assess the validity and fairness of the comparison.
> >
> > Second, while the adaptation to pixel space is understandable for image-based tasks, the feature-space formulation reflects a more general and flexible threat model, potentially applicable beyond images.
> > For completeness and to strengthen the generality of the claims, I encourage the authors also to consider evaluating robustness under feature-space attacks. I wish to know whether the proposed training method can maintain robustness in such settings as well.
> >
> > Lastly, I would appreciate a more precise justification for the focus on the $\ell_0$ pixel-level threat model. While sparsity in pixel perturbations is a conventional choice in image-based adversarial research, it is not immediately clear why this model would be more practical or impactful than, for example, an $\ell_1$-based threat model, which might better capture imperceptible sparse perturbations. If the assumed adversarial goal is to alter the input while remaining undetected minimally, then an $\ell_1$ attack could arguably be more appropriate.

---

> > > ### Comment · Reviewer_GeaB · 2025-08-02
> > >
> > > Furthermore, since it is now clear that the paper focuses on pixel-space $\ell_0$ attacks, it becomes even more important to compare against certified or provably robust methods developed for this specific setting.
> > > In particular, I encourage the authors to consider the works of Hammoudeh and Lowd [a] and Jia et al. [b], which offer theoretical robustness guarantees in the $\ell_0$  threat model and would serve as strong points of comparison for both robustness and practical performance.
> > >
> > > [a] Hammoudeh and Lowd Provable Robustness against a Union of $\ell_0$  Adversarial Attacks, AAAI 2024.
> > >
> > > [b] Jia et al. [b], Almost Tight $\ell_0$-Norm Certified Robustness of Top-k Predictions Against Adversarial Perturbations. ICLR 2022.

---

> ### Author Response · Authors · 2025-08-04
>
> Thanks for your detailed and insightful feedback. We would like to provide point-to-point responses to your questions as below and hope this can address your concerns.
>
> ---
>
> **1.Details of the adaptation of $\sigma$-zero to pixel space**
>
> **Reply:** $\sigma$-zero utilizes an approximated $l_0$-norm regularization term $\hat{l_0}=\sum_{i=1}^d\frac{x_i^2}{x_i^2+\sigma}$ ($\sigma>0$ is a smoothing hyperparameter) to constrain the sparsity of perturbations. However, it calculates the approximated $l_0$ norm in feature space. To calclate the pixel-space approximated $l_0$ norm,**we modify the regularization term to $\hat{l_0}=\sum_{i=1}^{hw}\frac{\sum_{j=1}^{c}x_{i,j}^2}{\sum_{j=1}^{c}x_{i,j}^2+\sigma}$**, where h, w and c are the height, width and channel of images, and $x_{i, j}$ represents the perturbation on the $j$-th channel of a particular pixel. As for the hyperparameters, **we adopt the ones designed for sAT/sTRADES models in the original paper**. Specifically, $\sigma=1$, $\tau_0=0.1$, where $\tau_0$ is the initial sparsity threshold used to guarantee full sparsity (clip the perturbation value lower than it to 0). The other hyperparameters are the same as the default ones.
>
> Moreover, we update the table above with the results of sPGD_p, sPGD_u and RS when they have the same query budget as $\sigma$-zero, i.e., 10000 queries, while sAA is the ensemble of sPGD_p, sPGD_u and RS. The observation is consistent with the previous one that $\sigma$-zero underperforms in pixel space.
>
> We admit that $\sigma$-zero is an effective white-box attack (in some particular cases, it overperforms white-box sPGD_p or sPGD_u) and will include it in our revised manuscript. However, sAA is still the most comprehensive and reliable toolkit to evaluate $l_0$ robustness, and models trained by our proposed algorithm can stably achieve competitive robustness very efficiently.
>
> |CIFAR-10|sAT|sAT+S&N|sTRADES|sTRADES+S&N|Fast-LS-$l_0$|
> |---|---|---|---|---|---|
> |Clean Acc.|84.5|80.8|89.8|82.8|82.5|
> |sPGD_p|75.9|76.8|84.6|74.1|67.2|
> |sPGD_u|75.3|75.1|81.7|72.2|67.7|
> |RS|36.2|61.1|61.8|66.1|65.4|
> |sAA|36.2|61.0|61.7|65.5|63.0|
> |$\sigma$-zero|79.8|78.7|85.9|77.0|73.7|
>
> |ImageNet-100|sAT|sAT+S&N|sTRADES|sTRADES+S&N|Fast-LS-$l_0$|
> |---|---|---|---|---|---|
> |Clean Acc.|86.2|83.0|84.8|82.4|82.4|
> |sPGD_p|78.0|78.8|80.6|78.2|74.6|
> |sPGD_u|77.8|79.2|81.4|79.8|74.6|
> |RS|61.4|75.0|76.0|78.2|76.8|
> |sAA|61.2|74.8|75.8|77.8|72.4|
> |$\sigma$-zero|78.6|80.8|81.6|80.0|74.0|
>
> ---
>
> **2. Robustness under feature-space attacks**
>
> **Reply:** Thanks for your constructive suggestion. We report the results of a model trained with our method in feature space below. To accommodate the feature-space sparsity, we modify the shape of perturbation mask in sPGD from (b,1,h,w) to (b,c,h,w). Since RS is unable to generate feature-space perturbations, we set its pixel-space adversarial budget to $\epsilon / c$ to ensure the equivalent feature-space budget as other attacks. **The results indicate that our method can also achieve robustness in this setting.** In addition, we notice that $\sigma$-zero outperforms sPGD in feature-space attacks, which is consistent with what the reviewer has pointed out. Nevertheless, sAA is still the most comprehensive and reliable robustness evaluation method.
>
> |Feature-space  $\epsilon=60$|Clean Acc.|sPGD_p|sPGD_u|RS|sAA|$\sigma$-zero|
> |---|---|---|---|---|---|---|
> |Fast-LS-$l_0$|84.9|76.6|73.5|58.5|58.3|64.2|
>
> ---
>
> **3. Effectiveness beyond images**
>
> To further validate the effectiveness beyond images, we employ our method on **tabular data**. We aim to perturb **one input feature** of DANet [7] trained on the forest cover type dataset (54 features in total). The training attack is 1-step sPGD ($\epsilon_{train} = 6$) and the evaluation attacks are 10000-step sPGD and $\sigma$-zero ($\epsilon = 1$). The results indicate that our method is still effective for tabular data. Due to the time limit, we cannot explore more application scenarios. Despite this, the results in Table 4, 7, 8, 10, and below can already demonstrate **the effectiveness of our method across different datasets, networks, and modalities**.
>
> |Model|Vanilla|1-step sAT|Fast-LS-$l_0$|
> |---|---|---|---|
> |Clean Acc.|93.9|88.5|86.4|
> |sPGD|0.2|31.5|45.8|
> |$\sigma$-zero|8.9|29.7|34.9|

---

> ### Author Response · Authors · 2025-08-04
>
> **4. Difference between $l_0$ pixel-space threat model and $l_1$ threat model**
>
> **Reply:** We would like to highlight that the **$l_1$-bounded perturbations are not strictly sparse [1]**. What is even worse, as indicated in [1], strong $l_1$ bounded adversarial attackers such as AA-l1 [2] tend to exploit non-sparse perturbations bounded by $l_1$ norms to successfully attack the model.
>
> Furthermore, **$l_0$ bounded sparse perturbations are quite common in the physical world**. Examples include dirt or sticker on a road sign, abnormal LED pixels and token replacement in a sentence. The sparse perturbations in these examples have their specific structures and can be better modeled by strict $l_0$ constraints instead of $l_1$ ones. Therefore, we believe the problems we study in this work have rich realistic applications, the methods we propose can be customized for broad downstream applications.
>
> Finally, **the optimization under the $l_0$-norm constraint is challenging** due to its non-convex nature. Methods for $l_1$, $l_2$ and $l_\infty$ bounded perturbations are usually invalid for $l_0$ perturbations. Therefore, research on the robustness against sparse adversarial attacks is also of great practical significance.
>
> ---
>
> **5. Certified robustness**
>
> **Reply:** Thanks for your suggestion. We only implement [b] since the source code for [a] is not available. However, we cannot certify the robustness of adversarially trained models, including the baselines in Table 5 and our methods, under the setting of [b]. The observation is consistent with $l_2$, $l_\infty$ cases. We provide our clarifications as follows:
>
> **(a)** There's a key distinction between **certified robustness** and **empirical robustness**, plain adversarial training usually only achieves the latter. Certified robustness establishes a lower bound on a model's true robustness, whereas empirical robustness establishes an upper bound on the true robustness. In this context, low certified robustness may arise from the certification algorithm's incapability to tightly bound the model's output. Extensive literature has indicated that the certification method needs to adapt to the training algorithm to achieve competitive performance; otherwise, it usually has trivial results. For example, the model trained by IBP [3] cannot be certified by CROWN [4] and vice versa. Specifically, models obtained by adversarially training, despite high empirical robustness, have almost zero certified robustness. **This phenomenon is broadly observed in $l_2$ and $l_\infty$ cases [3, 4, 5, 6]. The results in $l_0$ cases turn out consistent.**
>
> **(b)** **The setting of [b] is not applicable to adversarial training.** The analysis in [b] relies on **randomized smoothing**. For example, in the setting for CIFAR-10, the model should be trained on images with **974** randomly perturbed pixels, and theoretically, the certification algorithm in [b] can only certify robustness against **13** perturbed pixels in the most optimistic scenario (where the top-1 probability approaches 100%). Actually, the original paper only reports the certified robustness for at most 5 perturbed pixels, while our model is evaluated with an $\epsilon=20$. Based on the theoretical analyses in [b], **to certify robustness against 20 perturbed pixels, we need to perturb at least 990 pixels during training**, which dramatically affects the convergence speed and model utility.
>
> **(c)** As discussed in point (a), the certified robustness is very strict and establishes the lower bound of true robustness, in contrast to the empirical robustness, which our work focuses on. In addition, **the certification methods will dramatically increase the computational overhead**, either for the provable training (like CROWN [4], CROWN-IBP [5]) or for the Monte Carlo sampling during inference of a randomized model (like randomized smoothing [b] the reviewer mentioned). The computational overhead is contradictory to the motivation of this work: **boosting efficiency while maintaining (empirical) robustness.**
>
> ---
>
> [1] Towards stable and efficient adversarial training against l1 bounded adversarial attacks. ICML 2023.
>
> [2] Mind the Box: l1-APGD for Sparse Adversarial Attacks on Image Classifiers. ICML 2021.
>
> [3] On the Effectiveness of Interval Bound Propagation for Training Verifiably Robust Models. ICCV 2019.
>
> [4] Efficient Neural Network Robustness Certification with General Activation Functions. NeurIPS 2018.
>
> [5] Towards Stable and Efficient Training of Verifiably Robust Neural Networks. ICLR 2020.
>
> [6] Towards Fast Computation of Certified Robustness for ReLU Networks. ICML 2018.
>
> [7] DANETs: Deep Abstract Networks for Tabular Data Classification and Regression. AAAI 2022.

---

> > ### Comment · Reviewer_GeaB · 2025-08-05
> >
> > Dear authors,
> > I appreciate your responses and your commitment to this rebuttal. The discussion was stimulating and resolved my doubts. I will increase my score.

---

> > > ### Author Response · Authors · 2025-08-06
> > >
> > > We are happy that our responses addressed your concerns and really appreciate that you raised the rating. It is super inspiring for us. We will incorporate the critical points during the rebuttal in the revised paper. Thanks again for your commitment in reviewing our work.

---

### Official Review · Reviewer_pKXP · 2025-07-03

**Clarity:** 3
**Significance:** 2
**Originality:** 2
**Rating:** 4
**Confidence:** 3

**Summary:**

This paper conducts a systematic study of catastrophic overfitting (CO) in $l_0$ adversarial training. While regularization techniques such as soft labels and trade-off loss functions have been widely used to mitigate robust overfitting in $l_2$ and $l_\infty$ adversarial settings, this work focuses on analyzing the unique challenges presented by the $l_0$ case. The authors provide both theoretical analysis and extensive empirical results to demonstrate that smoothing the adversarial loss landscape through the use of soft labels and trade-off loss can effectively eliminate CO and significantly improve robustness against sparse $l_0$ attacks, with only a minor impact on standard accuracy. Importantly, the proposed smoothing techniques also substantially reduce the robustness gap between one-step and multi-step adversarial training, thereby making fast adversarial training much more practical and effective in the $l_0$ setting.

**Questions:**

1.	How sensitive is the performance to the choice of hyperparameters in the trade-off loss and label smoothing?

2.	The paper provides a detailed analysis of the differences between catastrophic overfitting in the $l_0$ setting and in the $l_2$/$l_\infty$ cases. Given these newly identified distinctions and the theoretical formulas presented, have the authors considered or explored any new or specifically tailored solutions for the $l_0$ case, beyond directly applying existing regularization methods? More dicussions are expected.

**Ethical Concerns:**

["NO or VERY MINOR ethics concerns only"]

**Final Justification:**

Thank the authors for the response which addressed many of my concerns. While I am still a bit unconvinced by the novelty, I think this paper is a good contribution to the community. I will keep my positive rating.

**Limitations:**

YES

**Quality:**

3

**Strengths And Weaknesses:**

Strengths:

•The paper addresses an underexplored and practically important problem: catastrophic overfitting(CO) in $l_0$ adversarial training.

•Theoretical analysis offers new insights about the discrete and highly non-smooth nature of the $l_0$ loss landscape.

•Experiments are thorough and well-controlled, reporting both robust and clean accuracy, and ablation studies confirm the effectiveness of regularization techniques.

Weaknesses:

•The techniques (soft label, trade-off loss) appear  not new and have been extensively studied in the context of $l_2$/$l_\infty$ adversarial training.

•The main contribution is a systematic analysis and empirical demonstration of these techniques’ value in the $l_0$ setting, rather than proposing fundamentally new methods.

•Some practical aspects, such as hyperparameter sensitivity, could be discussed in more detail.

---

> ### Author Rebuttal · Authors · 2025-07-28
>
> Thanks for acknowledging that the theoretical analysis is insightful and the experiments are thorough and well-controlled. We provide point-to-point responses to your concerns as follows:
>
> ---
>
> **W1: The techniques appear not new and have been extensively studied.**
>
> **Reply:**  We clarify that the rationale for using soft labels and trade-off loss functions is different for $l_0$ cases, since we point out that existing CO mitigation methods for $l_1$, $l_2$ and $l_\infty$ cases turn out to be insufficient to address the CO issue in $l_0$ cases. Smoothing the loss function can help boost the performance for $l_1$, $l_2$ and $l_\infty$ cases, as the goal of soft label and trade-off loss function is to address robust overfitting to boost performance. By contrast, smoothing the loss function is **essential** to address the CO issue in $l_0$ cases.
>
> ---
>
> **W2: The main contribution is a systematic analysis and empirical demonstration of these techniques’ value in the l_0 setting, rather than proposing fundamentally new methods**
>
> **Reply:** The main contribution of this work is **highlighting the challenge of fast $l_0$ adversarial training (AT) is different from the $l_1$, $l_2$ and $l_\infty$ counterparts**, specifically the suboptimal perturbation location. **To the best of our knowledge, this is a novel finding**. Although the multi-$\epsilon$ strategy can help alleviate this suboptimality by perturbing more pixels, it causes a craggy loss landscape. In this context, soft labels and trade-off loss functions are proposed to smooth the loss landscape. However, we acknowledge that our solution is mostly intuitive and may have room for improvement. We will leave further methodology improvement as a future work.
>
> ---
>
> **W3: Some practical aspects, such as hyperparameter sensitivity, could be discussed in more detail.**
>
> **Reply:**  We have conducted a comprehensive ablation study on hyperparameters in Appendix F.10, the results indicate that our method is robust to different hyperparameters. Results in Table 13, 14, 15 and 16 indicate that our method can always achieve competitive performance, similar to the multi-step counterpart, as long as the hyper-parameters are set reasonable values. In addition, no case in these results in ablation study suffers from CO, validating the stability of our method.
>
> ---
>
> **Q1: How sensitive is the performance to the choice of hyperparameters in the trade-off loss and label smoothing?**
>
> **Reply:** Please see the response to W3.
>
> ---
>
> **Q2: Have the authors considered or explored any new or specifically tailored solutions for the l_0 case, beyond directly applying existing regularization methods?**
>
> **Reply:** In this work, we mainly focus on exploring the common solution to mitigate CO in fast $l_0$ adversarial training, and find the combination of TRADES and SAT, which improves both 1st and 2nd Lipschitz smoothness, performs the best. Despite intuitive, extensive experiments validate the effectiveness of our method in various tasks and models. **We believe that the motivation of smoothing loss landscape and our exemplary methods can be broadly applied in the situations where we need to efficiently robustify the model against a kind of sparse perturbations**. Nevertheless, we acknowledge the necessity to develop specifically tailored method for the $l_0$ case to further boost the performance and will leave it as the future work.
>
> ---
>
> **We hope these responses can address your concerns and look forward to your feedback.**

---

> ### Author Response · Authors · 2025-08-06
>
> We are happy that our responses addressed many of your concerns and acknowledge our work's contribution. We will incorporate the critical points during the rebuttal in the revised paper. Thanks again for your commitment in reviewing our work.

---

### Note · Authors · 2025-08-12

We appreciate the effort of AC and reviewers during the rebuttal. We want to highlight the contributions of this work as follows:

1. We highlight that catastrophic overfitting (CO) in fast l0 adversarial training arises from **sub-optimal perturbation locations** in 1-step attacks. Techniques effective in $l_\infty$, $l_2$, and $l_1$ cases cannot fully address the CO issue in the $l_0$ case.
2. Our theoretical and empirical analysis shows that **the adversarial loss landscape is more craggy in $l_0$ cases, exacerbating CO**. In this regard, we can adopt **soft labels** and the **trade-off loss function** to provably smooth the adversarial loss landscape.
3. **Comprehensive experiments** demonstrate that smoothing the loss function significantly narrows the performance gap between 1-step $l_0$ adversarial training and its multi-step counterparts, achieving state-of-the-art performance in efficient adversarial training against sparse perturbations.

In the rebuttal, we addressed the following main concerns of reviewers:

1. Report the robust accuracy against $\sigma$-zero, demonstrating that our methd is still robust to $\sigma$-zero and sAA is a more effective attack for our models. (**GeaB**)
2. Demonstrate the effectiveness of our method in feature-space perturbations and other modalities. (**GeaB**)
3. Clarify some details of this work.

---

### Decision · Program_Chairs · 2025-09-17

**Decision:**

Accept (poster)

**Comment:**

Thanks for submitting to NeurIPS 2025. The paper studies the ineffectiveness of 1-step adversarial training against $l_0$ bounded perturbations. The craggy loss landscape of $l_0$ adversarial training is identified as the major cause. The paper then proposes soft labels and trade-off loss function to smooth the landscape and achieve strong robustness against $l_0$ perturbations within the 1-step adversarial training scheme. Reviewers are overall positive about the paper, especially after the discussion phase where authors put a diligent effort to address concerns on experimental evaluation and theoretical foundations. After the discussion phase, all reviewers are positive. There are some minor concerns about novelty but not critical. We recommend acceptance. Congratulations! Authors should incorporate the reviews and responses in the camera-ready version.